# Can local climate variability be explained by weather patterns? A multi-station evaluation for the Rhine basin

Aline Murawski[1], Gerd Bürger[2, 3], Sergiy Vorogushyn[1], and Bruno Merz[1]

[1]GFZ German Research Centre for Geosciences, Potsdam, Germany
[2]Institute of Meteorology, FU Berlin, Germany
[3]Institute of Earth and Environmental Science, University of Potsdam, Germany

*Correspondence to:* Aline Murawski (murawski@gfz-potsdam.de)

**Abstract.** For understanding past flood changes in the Rhine catchment and in particular the role of anthropogenic climate change for extreme flows, an attribution study relying on a proper GCM (General Circulation Model) downscaling is needed. A downscaling based on conditioning a stochastic weather generator on weather patterns is a promising approach. This approach assumes a strong link between weather patterns and local climate, and sufficient GCM skill in reproducing weather pattern climatology. These presuppositions are unprecedentedly evaluated here using 111 years of daily climate data from 490 stations in the Rhine basin and comprehensively testing the number of classification parameters and GCM weather pattern characteristics. A classification based on a combination of mean sea level pressure, temperature, and humidity from the ERA20C reanalysis of atmospheric fields over Central Europe with 40 weather types was found the most appropriate to stratify six local climate variables. The corresponding skill is quite diverse though, ranging from good for radiation to poor for precipitation. Especially for the latter it was apparent that pressure fields alone cannot sufficiently stratify local variability. To test the skill of the latest generation of GCMs from the CMIP5 ensemble in reproducing the frequency, seasonality, and persistence of the derived weather patterns, output from 15 GCMs is evaluated. Most GCMs are able to capture these characteristics well, but some models showed consistent deviations in all three evaluation criteria and should be excluded from further attribution analysis.

## 1 Introduction

The Rhine River is a trans-boundary river with a catchment area of $185\,000\,\mathrm{km}^2$ and significant flood risk. Along the main river reach from Karlsruhe in south-west Germany to Rees at the Dutch-German border, an area of $14\,600\,\mathrm{km}^2$ is at risk of being flooded for an extreme scenario with a return period of 200 to 500 years (Thieken et al., 2015). This enormous economic exposure to floods is accompanied by expectations that flood magnitudes will increase due to climate change (e.g. Dankers and Feyen, 2009; te Linde et al., 2010; Bosshard et al., 2014). Further, the Rhine catchment has experienced increasing flood trends during the second half of the 20th century (Petrow and Merz, 2009). It has been argued that climatic drivers, land use changes and river training may have contributed to the observed trends (Pinter et al., 2006; Petrow et al., 2009; Villarini et al., 2011; Vorogushyn and Merz, 2013). Whereas the role of river training in the main Rhine channel has been quantified (Lammersen et al., 2002; Vorogushyn and Merz, 2013), the effect of climatic and land use changes remains unclear. In particular, the

contribution of anthropogenic climate change on flood trends is an open question. To understand the role of climatic drivers for past changes in river flooding, rigorous attribution studies are needed (Merz et al., 2012).

Several studies tried to quantify the role of changes in meteorological variables on river flows using hydrological models with alternative sets of climate drivers (Hamlet and Lettenmaier, 2007; Hamlet et al., 2007; Hundecha and Merz, 2012; Duethmann et al., 2015). If an attribution of hydrological changes to changes in the atmospheric composition such as greenhouse gas concentration is attempted, output from GCMs (General Circulation Models), representing two different "worlds" with and without anthropogenically induced climate change, are to be compared (Min et al., 2011). This requires that output of GCMs is properly downscaled to a resolution compatible with hydrological models.

Different approaches are applied in the hydrological community for statistical downscaling (for a review see Fowler et al., 2007; Maraun et al., 2010). Statistical downscaling approaches using weather generators offer the possibility to generate multiple realisations of long synthetic time series, e.g. 100 years of daily values, and are considered to have similar skills compared to RCMs (Hewitson and Crane, 2006). This provides a basis for a more robust estimation of changes of hydrological variables and moments of their distributions. They are particularly suited for quantifying rare floods and their impacts (e.g. Falter et al., 2015), in case they are capable to represent the statistical behaviour of extreme events. Examples of using weather generators to bridge the spatial gap between GCMs and hydrological impacts are widespread (e.g. Wilks, 1992; Katz, 1996; Semenov and Barrow, 1997; Fowler et al., 2000, 2005; Elshamy et al., 2006; Hewitson and Crane, 2006; Kilsby et al., 2007; te Linde et al., 2010; Fatichi et al., 2011; Lu et al., 2015; Kim et al., 2015)

In order to represent different climate states, parameters of a weather generator can be conditioned on the climate model output by applying a change factor (Kilsby et al., 2007) or on covariates such as weather patterns. The latter approach is expected to better capture change in variability of the changing climate state. Weather patterns are classifications of atmospheric circulation fields or other synoptic fields (Huth et al., 2008). The underlying assumption of the downscaling based on weather patterns is that the regional or local behaviour of climate variables is partly a response to the larger-scale, synoptic forcing. The weather generator is then parameterised separately for each class of weather patterns (e.g. Bárdossy and Plate, 1991, 1992; Corte-Real et al., 1999; Fowler et al., 2005; Haberlandt et al., 2015). Statistical downscaling tends to underestimate the variance of regional or local climate if the contribution of local processes is not considered and may poorly represent extremes. Different methods have been proposed to rectify this problem: variable inflation (Karl et al., 1990), expanded downscaling (Bürger, 1996) and randomisation (Kilsby et al., 1998). This problem typically occurs in downscaling approaches that are based on regression models and weather patterns. It is circumvented when a weather generator is conditioned on weather patterns, provided that the weather generator is able to adequately capture the tail behaviour of the surface climate variables.

A downscaling approach based on weather pattern classification builds on four assumptions. Firstly, local climate needs to be sufficiently explained by the classification of the large-scale synoptic situation. Bárdossy et al. (2002) summarise that many studies have shown that there is a strong link between atmospheric circulation types derived from CTCs (circulation type classifications) and surface variables such as near-surface temperature and precipitation. Even when the small-scale climate is governed by mesoscale events such as convective systems, these are, in turn, conditional on the synoptic state (Goodess and Jones, 2002). On the other hand, weather patterns can only be a proxy for local weather, due to the categorisation of continuous

data by the discrete classification, and more importantly, due to the fact that the large-scale situations do not fully represent smaller-scale features. This so-called within-type variability (e.g. Huth et al., 2008) is caused, for instance, by small-scale processes, such as orography-enhanced rainfall, or by variations in dynamic properties (pressure gradient, vorticity, intensity) of weather patterns (Beck et al., 2007).

Secondly, the linkage between weather patterns and regional and local climate is assumed to be stationary. This means that climate change will mainly manifest itself as a change in the frequency, persistence and seasonality of these weather patterns. The transfer function between synoptic state and regional and local climate thus remains constant. Land use and land cover change, for example, could introduce a variable forcing on local climates (Hewitson and Crane, 2006). Using long observational time series, it has been argued that the linkage between large-scale weather patterns and regional climates are characterised by

distinct variabilities (Beck et al., 2007).

The third assumption is that GCMs are able to properly reproduce weather patterns. GCMs are often strongly biased in variables such as precipitation (e.g. Sunyer et al., 2015), but are expected to reflect large-scale circulations well. This skill in representing the synoptic situations compared to the poor skill in representing surface variables is utilised for statistical downscaling. For example, Hewitson and Crane (2006) conclude that much of the discrepancy between GCM projections of

precipitation over South Africa may result from differences in their precipitation parametrisation schemes, whereas the synoptic dynamics are well simulated. It has been shown, however, that the skill of GCMs to reproduce weather pattern characteristics such as geopotential height, sea level pressure etc. varies strongly (Brands et al., 2013; Wójcik, 2015).

Finally, to obtain meaningful input for the hydrological model, a weather generator has to adequately represent the space-time dynamics of the catchment meteorology. This is a particular challenge for large river basins, where the correlation structure

of e.g. precipitation becomes difficult to capture over large distances.

In the presented paper we evaluate the assumptions for weather pattern based downscaling for the Rhine catchment. This is a prerequisite for conditioning a weather generator on circulation patterns for understanding the role of climatic drivers for past and future flood changes in the Rhine basin. We focus on the first and third assumption here. The assumption of stationarity of the linkage between weather patterns and local climate and the skill of the weather generator itself will be investigated

separately. In the future we intend to use a multi-site, multi-variate weather generator (Hundecha et al., 2009; Hundecha and Merz, 2012) for downscaling GCM output to drive a regional hydrological model. Extreme value statistic on the simulated streamflow will then allow to quantify the role of climatic change on flood flows.

To underpin the first and third assumptions, we derive an "optimal" weather pattern classification and investigate (1) to which extent weather patterns are able to stratify local climate variables, and (2) the skill of the GCMs to reproduce these weather

patterns. It has been argued that there is no "best" statistical downscaling approach but that the optimal classification depends on the application and region (Hewitson and Crane, 2006; Huth et al., 2008). We look specifically from the perspective of a hydrological impact study for the Rhine catchment.

There is a significant body of literature on weather pattern classification. Our work extends these studies in several aspects. Firstly, we test the skill of several classification variables. Often classifications are based on msl (mean sea level pressure)

only. We use, in addition, the synoptic temperature and humidity fields to classify weather patterns. Considering temperature

as classification variable has the advantage that one classification can be used throughout the year. Secondly, we test the ability of weather pattern classifications to stratify a comparatively large number of climate variables with daily resolution: precipitation, minimum, mean, and maximum temperature, radiation, and relative humidity. Other studies often consider only one or two variables (e.g. Beck et al., 2007; Kyselý, 2007; Anagnostopoulou et al., 2008; Beck and Philipp, 2010; Łupikasza, 2010; Haberlandt et al., 2015) and only few studies are available with an extended list of up to eight variables (e.g. Kidson, 1994; Enke et al., 2005a; Cahynová and Huth, 2010). We use a comparatively long time period of 111 years. The periods of other studies are typically much shorter, e.g. 11 to 50 years (Kidson, 1994; Brinkmann, 1999, 2000; Goodess and Jones, 2002; Hewitson and Crane, 2006; Anagnostopoulou et al., 2008; Beck and Philipp, 2010; Brisson et al., 2010; Łupikasza, 2010; Cahynová and Huth, 2010; Bettolli and Penalba, 2012), or 100 years (Kyselý, 2007). Beck et al. (2007) covers a longer period, going back to 1780, however using only monthly resolution. Fourthly, our analysis covers a large, transboundary area of around $160\,000\,\text{km}^2$ and a very large number of climate stations (490). Weather pattern classifications typically work with a comparatively low number of stations, ranging from e.g. one station for Prague (Kyselý, 2007) to 84 stations for New Zealand (Kidson, 1994).

Further, we analyse the newest generation of climate models from the Coupled Model Intercomparison Project Phase 5 (CMIP5). We investigate their ability to reproduce frequency, persistence, and seasonality of weather patterns. Wójcik (2015) emphasises the need to assess the reliability of GCMs prior to any statistical downscaling approach. Whereas Perez et al. (2014) analysed the frequency of patterns over the north-east Atlantic and Belleflamme et al. (2013) examined frequency and persistence of patterns over Greenland, so far no other study analysed seasonality as done here. Particularly for understanding the role of climate change on flood flows, matching the seasonality is essential.

## 2   Data

For the workflow proposed here, three different sets of climate data are needed: (1) data to establish the weather pattern classification on, (2) compatible output of climate models with different greenhouse gas (GHG) forcings, i.e. same variables and spatial coverage as (1), (3) observations from local climate stations in the investigated area (Rhine catchment) for all meteorological variables of interest, preferably covering the same time period as (1).

To investigate the suitability of different climate variables to establish the weather pattern classification, long-term reanalysis fields can be used. We utilised the newly available ERA-20C – a gridded reanalysis data set from the ERA-CLIM project (Poli et al., 2013). This data set is a pilot reanalysis of the 20th-century assimilating surface observations only and being forced by a HadISST2.1.0.0 ensemble of sea-surface temperature and sea-ice conditions, available for the period 1900–2010. The 3- or 6-hourly data, depending on the variable, were aggregated to daily averages for this study. The spatial resolution of $1° \times 1°$ was chosen. There are finer resolutions available for ERA-20C, but the resolution of GCMs is not finer than $1.25° \times 0.94°$.

The skill of different weather pattern classifications was assessed according to their ability to stratify climate station data located in the Rhine catchment (Figure 1). Sets of daily precipitation, temperature (mean, min, max), relative humidity, and global radiation data for the period 1901–2010 were available from the national meteorological services and kindly processed

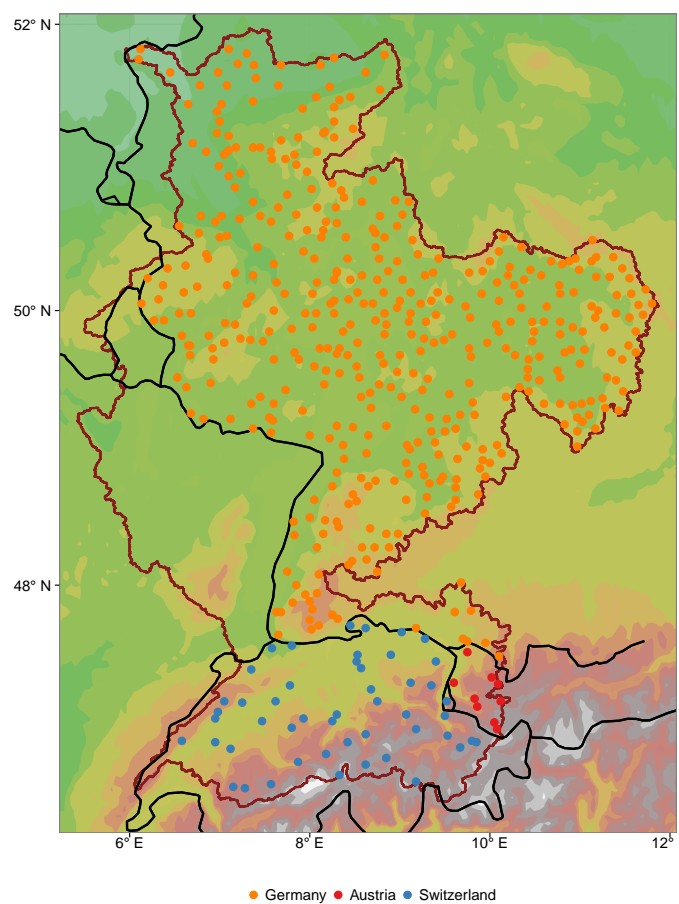

● Germany ● Austria ● Switzerland

**Figure 1.** Locations of climate data stations used. See text for more details on single data sets. Dark red line shows Rhine catchment, black lines denote state borders.

and quality controlled by the Potsdam-Institute for Climate Impact Research (PIK) (Österle et al., 2006). For the German part of the catchment 432 stations were available, nine stations for the Austrian part, and 49 stations for Switzerland and Liechtenstein. To date no data of meteorological stations in France were available. This set of 490 climate stations allows for the classification results to be compared to a large and dense station network.

For the assessment of the effect of anthropogenic GHG emissions on changes in floods, data from modelling experiments with two different GHG forcings representing (a) the historical (natural+anthropogenic) GHG concentrations (All-Hist) and (b) only natural GHG concentrations (Nat-Hist) are required. These experiments are available from a number of GCMs of the CMIP5 project (Taylor et al., 2012). An overview on the models and the number of runs available for the All-Hist experiment used here is given in Table 1. The model output is available in daily time steps, mostly starting as early as the mid-19th

century. All available runs were analysed in relation to the ability of different GCMs to replicate the frequency, persistence, and seasonality of weather patterns.

# 3 Methods

## 3.1 Weather pattern classification

Within the COST Action 733 "Harmonisation and Applications of Weather Type Classifications of European Regions" a collection of circulation type classification approaches was compiled and made available (cost733class software: http://cost733. geo.uni-augsburg.de/cost733class-1.2/, Philipp et al., 2016). Included, among others, is the SANDRA classification method (simulated annealing and diversified randomisation) which is "a non-hierarchical technique for minimising the sum of Euclidean Distances within the classes" (Philipp, 2009). The method is similar to k-means clustering, but is able to get closer to

the global optimum instead of getting trapped in a local one. A detailed description of the method can be found in Philipp et al. (2007). Several studies found a good or even superior performance of SANDRA compared to other classification methods (e.g. Beck and Philipp, 2010; Huth, 2010; Huth et al., 2008; Philipp, 2009; Philipp et al., 2016, 2007).

Available in the cost733class software are also methods for assigning new data to an already existing classification. This was used to apply the selected classification to GCM data. The method takes data of the same spatial domain and resolution and

compares every case, i.e. day, to the centroids of the existing classification. The class with the smallest Euclidean Distance to the respective case is assigned. In this way a catalogue (i.e. time series) of weather patterns can be obtained for every GCM data set, which can then be analysed and compared to the catalogue derived from reanalysis data (see subsection 4.2). Since the GCMs do not necessarily have the same spatial resolution as the classification input, they were first linearly re-interpolated to the same grid as the ERA-20C data.

By employing a weather pattern classification we are aiming towards providing a stratification of observed weather variables such as precipitation, temperature, relative humidity and solar radiation (as required for the hydrological model). For use with the weather generator, it is desired to obtain a classification that provides patterns that are preferably as distinct as possible from each other in terms of local weather characteristics. To derive an optimal classification, different characteristic variables, e.g. msl, geopotential height, temperature, humidity, different spatial domains and different numbers of weather type classes

can be tested. Historically, first classification were based on sea level pressure. An improvement of classifications and variable stratification can be achieved by additionally considering geopotential height (Spekat et al., 2010; Nied et al., 2014). Given the further aim of this classification to be used for downscaling of historical runs of CMIP5 models, geopotential height is available only in a few runs and is thus excluded from our consideration.

Note that the term "weather (pattern) classification" is used to contrast the difference to air mass classifications, since surface

weather variables are used here instead of variables defined at different tropospheric levels (Huth et al., 2008).

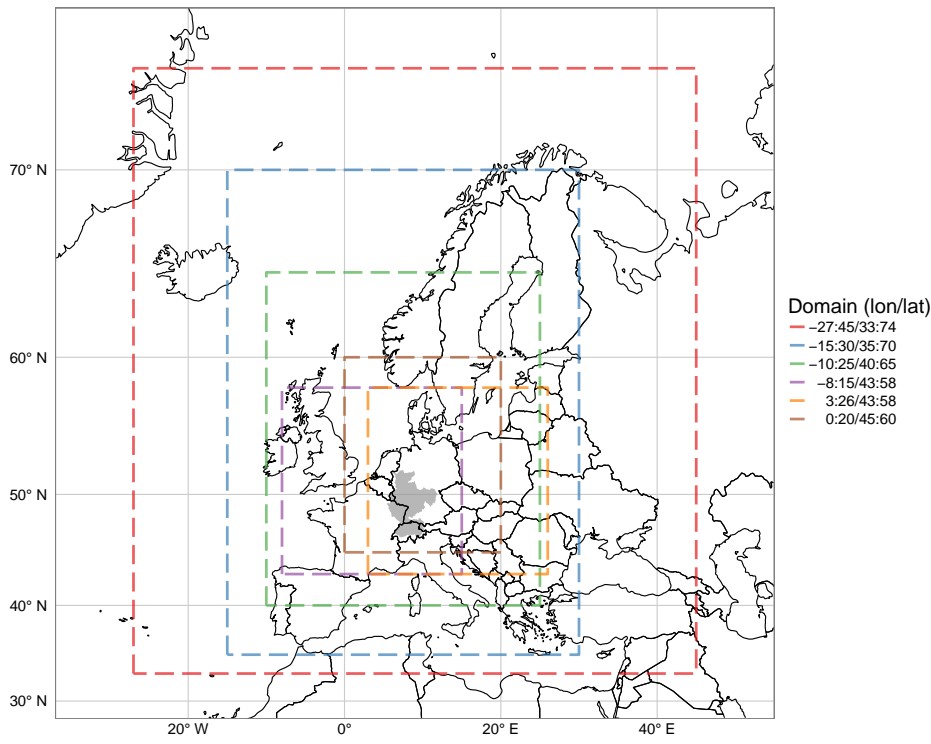

**Figure 2.** Spatial domains of weather pattern classifications in degree of geographic longitude/latitude. Dark grey polygon shows location of Rhine catchment. Domain 3 : 26 / 43 : 58 as in Philipp et al. (2010), region D07; -15 : 30 / 35 : 70 as in Nied et al. (2014).

### 3.2 Finding optimal classification parameters

Here we tested different combinations of variables for weather type classification. Classifications on mean sea level pressure (msl) are commonly applied (e.g. Philipp, 2009; Wilby and Quinn, 2013; Masson and Frei, 2014). Other frequently used variables include geopotential height of different levels, thickness between different levels, vorticity and temperature at certain levels, or total column water vapour (e.g. Bárdossy et al., 2002; Anagnostopoulou et al., 2008; Nied et al., 2014; Philipp et al., 2016). However our selection was restricted to variables that are also available from the GCM outputs. Goodess and Jones (2002) state that temperature and humidity are the two most important variables to be included when using a circulation-type approach for downscaling of rainfall. Thus we included temperature in 2 m (temp) (used, among other variables, in e.g. Kalkstein et al., 1987) and specific humidity (hus, as e.g. in Hewitson and Crane, 2006). This led to four combinations of variables: msl, temp, msl+temp, msl+temp+hus.

Different options for the selection of a spatial domain were tested here: one covering the whole of Europe, others being considerably smaller, partly focussing on the Rhine catchment, see Figure 2. One domain is identical to domain D07 in Philipp et al. (2010), another one is a westward shifted version of it. The domain from Nied et al. (2014) was included as well.

10  A wide range of number of classes was tested to assess the power of classification: 9, 18, 27 (all frequently used, e.g. in Philipp et al., 2010; Huth et al., 2016), 40 (as in Nied et al., 2014; Philipp, 2009; Bissolli and Dittmann, 2001). Many authors (e.g. Huth, 2010) consider 40 already a very large number, but e.g. Jones and Lister (2009) use 6–11 patterns per season, in total 34. Thus, when establishing a classification for the whole year a greater number of classes can be useful.

These different parameter sets allow for 120 possible combinations which poses an intractable computational problem. To break this number down in a reasonable way that still yields reliable results, firstly, some parameter values were prioritised (domains (lon/lat in degrees) $-27:45/33:74$ and $-8:15/43:58$, 18 and 40 classes). Secondly, four classification variables were combined with four prioritised parameters and the best-performing variable (combination) was selected. This variable

5  was then combined with all spatial domains finding the optimal one. Finally, all number of classes were evaluated with the best variable and domain. This reduces the number of combinations to 26, which is still a rather large computational effort.

### 3.3   Evaluation of classifications

First of all it has to be clear, if the classification itself should be evaluated (i.e. stratification of the input variables, such as msl) or if the stratification of other variables, such as precipitation, that were observed on days with certain weather patterns should be evaluated based on the developed classification. The latter is needed here. Hence, given a certain classification catalogue, data from weather stations can be assigned to the patterns that occurred at the same day, resulting in a distribution of values associated with each pattern. The distribution of values linked to a pattern can then be compared to the original (complete)

population of values.

The quality of a given classification can be evaluated using different statistical metrics. For example, Huth et al. (2008) and Beck and Philipp (2010) give various quality measures, among them the Explained Variation (EV), and the so-called Pseudo-F statistic (PF). These are chosen, because EV is frequently used in similar applications and is easily understood, while PF has the advantage of considering the number of classes and cases per class.

The Explained Variation (Equation 1) is defined as the ratio of the sum of squared deviations from the mean within classes (WSS) and the total sum of squared deviations from the overall mean (TSS). In Equation 3 and Equation 4 $k$ denotes the number of classes (i.e. patterns), $m$ is the number of dimensions (i.e. variables), $n$ is the number of cases (i.e. days), and $C_j$ denotes class/pattern $j$. Thus EV ranges between zero (poor) and one (perfect stratification).

The Pseudo-F statistic (PF, Equation 2) of Caliński and Harabasz (1974) is the ratio between the sum of squared deviations

between means of classes (BSS, Equation 5) and the sum of squared deviations within classes (WSS, Equation 4), weighted by the number of classes and cases. A minimum of within-type variation (and maximum of distinction between types/classes) is achieved by large values of PF, poor clustering is denoted by values close to zero.

Both indices are usually applied to one meteorological variable at a time, thus evaluating the skill of the classification in stratifying e.g. temperature or precipitation (Huth et al., 2016). When mapping each variable per weather pattern, it becomes

evident that some patterns might be very similar with regards to one (or more) variable(s), while being substantially different in other variables. For example in Figure 3, the selected patterns no. 12, 14, and 33 have a very similar mean temperature for

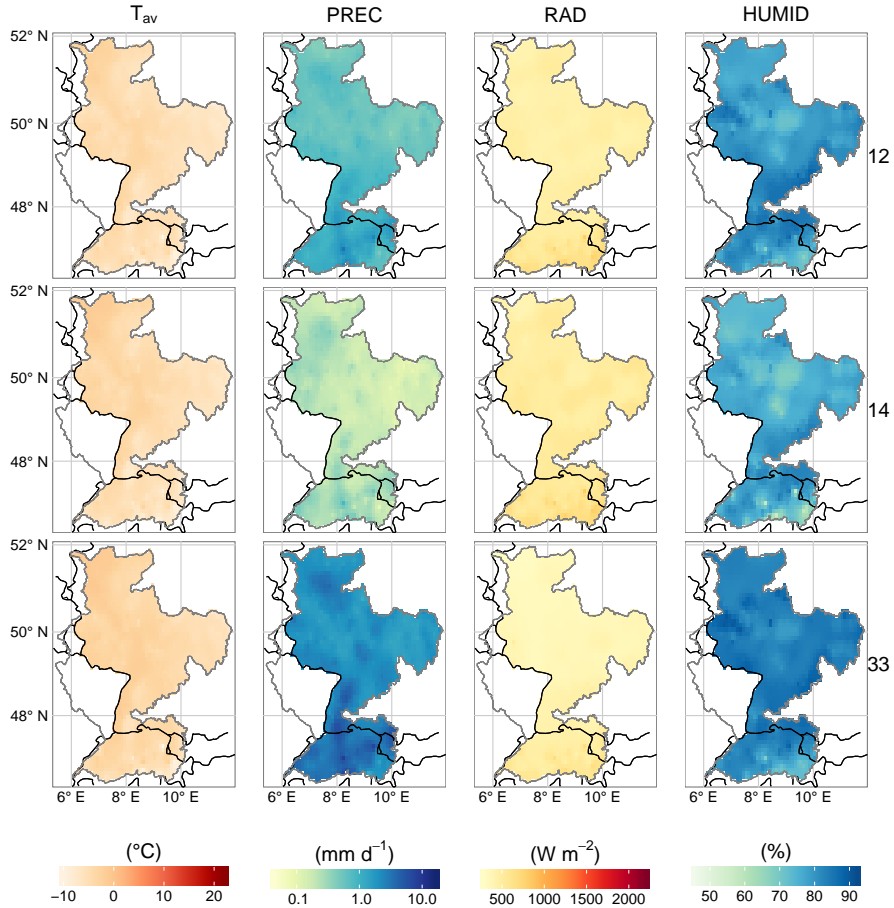

**Figure 3.** Average daily values of meteorological variables for example patterns 12, 14, and 33 to emphasise the need of multi-variate evaluation of weather pattern classifications ($T_{av}$ – average temperature, PREC – precipitation, RAD – global radiation, HUMID – relative humidity). Black line show state borders, grey outline denotes Rhine catchment.

the whole area but very different precipitation. A classification focussing only on one variable would neglect the variability of the others. We therefore evaluate the stratification with respect to both single- and multi-variate performance.

Each evaluation metric is applied to normalised climate data, derived separately for each station and aggregated as an area-weighted average over the complete Rhine catchment.

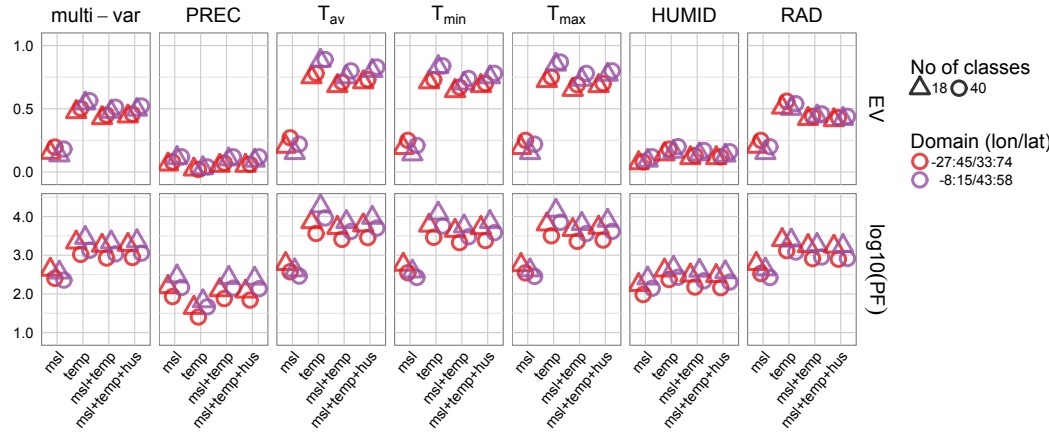

**Figure 4.** Evaluation metrics for the selection of classification variables (x-axis). Weather variables from station data in columns. Note log scaling of PF.

$$EV = 1 - \frac{WSS}{TSS} = \frac{BSS}{TSS} \tag{1}$$

$$PF = \frac{BSS/(k-1)}{WSS/(n-1)} \tag{2}$$

$$TSS = \sum_{i=1}^{n} \sum_{l=1}^{m} (x_{il} - \bar{x}_l)^2 \tag{3}$$

$$WSS = \sum_{j=1}^{k} \sum_{i \in C_j} \sum_{l=1}^{m} (x_{il} - \bar{x_{jl}})^2 \tag{4}$$

$$BSS = \sum_{j=1}^{k} n_j \sum_{l=1}^{m} (\bar{x_{jl}} - \bar{x}_l)^2 \tag{5}$$

## 4 Results

### 4.1 Stratification of local climate variables

#### 4.1.1 Selection of classification variables

For selecting the classification variables, both evaluation metrics (EV and PF) point to the same choice (see Figure 4). The multi-variate evaluation clearly suggests a classification including temperature (EV around 0.5). This preference is even

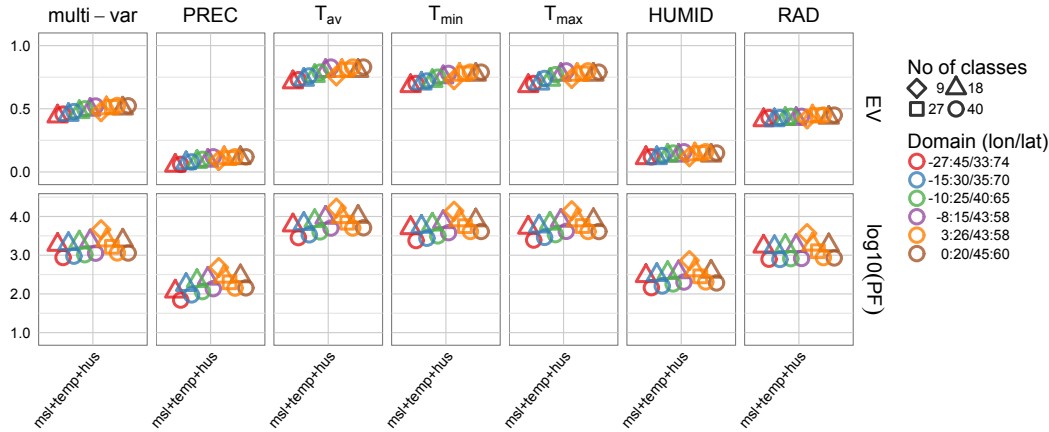

**Figure 5.** Evaluation metrics for the chosen classification variables, combining results of selection steps 2 and 3. Aims at selecting (a) best spatial domain (colour scale), then (b) selecting best number of classes (point shapes).

stronger for single-variate evaluations of temperature ($T_{av}$, $T_{min}$, $T_{max}$) with explained variation (EV) around 0.75. For precipitation (PREC) the temp-only classification performs worst, though EV values are low for all classifications (EV < 0.2). From the literature there is no evidence that other studies acquire considerably better results in similar analyses, but surpris-

ingly the exact values of their evaluation criteria are typically not given. Nevertheless this low skill needs to be discussed further (see also subsubsection 4.1.3 and 5). Any classification including msl improves the stratification of precipitation compared to the classification based on temperature only. Thus a classification including both, temperature and mean sea level pressure should be chosen to obtain a reasonably good stratification of all variables.

For relative humidity (HUMID) and global radiation (RAD) the same relation between classifications as for temperature

was found (classification including temp better than msl only), although the differences between classifications for HUMID are small. Including specific humidity as classification variable slightly improves the stratification of all variables. Thus the classification on msl+temp+hus was finally selected. This selection holds the additional advantage of a strong seasonal restriction of pattern occurrence. While patterns from a msl-only classification show only weak seasonality (i.e. each patterns might occur in any month throughout the year), the use of raw values (i.e. no anomalies) of temperature and specific humidity con-

fines each pattern to a specific season with a clear peak of occurrence in a certain month. This allows to use one classification for the whole year instead of using separate classifications for each season, as frequently done in other studies.

For both metrics and all meteorological variables the smaller spatial domains deliver better results (Figure 5). The three smallest domains (coloured in purple, orange, yellow) differ only in their exact location, but are of roughly the same size. The orange domain, gives slightly better results for all variables and was chosen for further analysis.

The choice of an optimal number of classes is less obvious (Figure 5). The analysis of the EV shows a slight tendency for a greater number of classes, whereas PF prefers a lower number. However, for the use with a weather generator, a high number of classes with consequently narrow distributions for each class are preferred. At the same time a sufficient amount

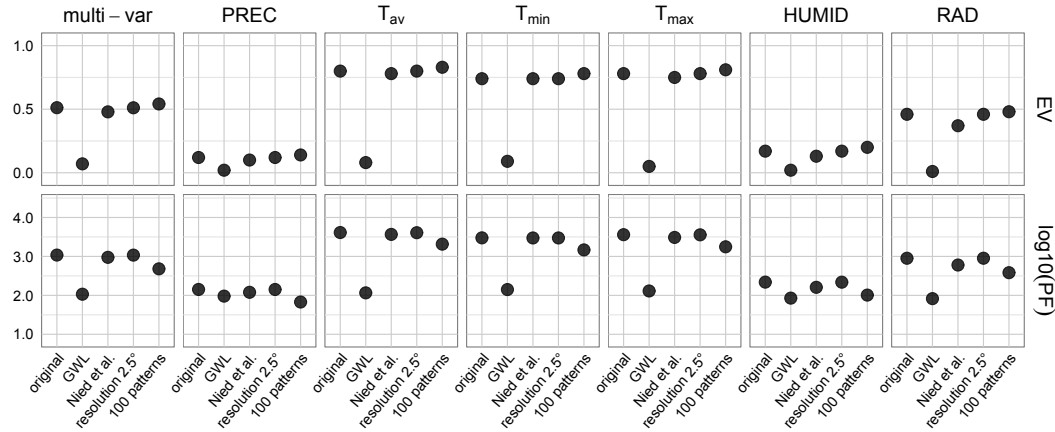

**Figure 6.** Comparison of selected classification from subsubsection 4.1.1 (original) and other classifications: Hess-Brezowsky-Grosswetterlagen (GWL), classification variables as in Nied et al. (2014), a classification on a coarse grid, and one with 100 classes.

of observations per class are needed for fitting the distributions. Considering this tradeoff a classification with 40 classes was selected here.

Average values of meteorological variables per pattern of the final classification are shown in Appendix A Figure 11–16.

### 4.1.2 Comparison to other classifications

The selected classification was compared with the Hess-Brezowsky-Grosswetterlagen (GWL) catalogue of circulation patterns, to the classification after Nied et al. (2014), and to two experiments where only one parameter of the selected classifcation was modified (Figure 6): A classification based on a coarse grid ($2.5° \times 2.5°$ instead of $1° \times 1°$), and one using 100 classes (as in Perez et al., 2014). A comparison to the well-established Hess-Brezowsky-Grosswetterlagen (GWL) (applied in e.g. Kyselý, 2007; Fleig et al., 2015) shows that GWL performs inferior to our classification with EV values not exceeding 0.1. The

stratification skill obtained by GWL is best comparable to a classification based on msl only, but is inferior when including other variables into the classification scheme. The classification based on $500\,\mathrm{hPa}$ geopotential height, $500\,\mathrm{hPa}$ temperature and total column water vapour as used by Nied et al. (2014) performs equally well as the selected classification with only slightly lower skill values.

ERA-20C data were originally used with $1° \times 1°$ resolution. A coarser resolution of $2.5° \times 2.5°$ results in an identically good

stratification. Hence small-scale features that might be present in a high-resolution reanalysis data set do not distort the results, which is also true for a classification extent covering all of Europe (not shown here).

A last test was dedicated to the number of patterns: 100 patterns as in Perez et al. (2014) were tested, confirming the general tendency (increasing EV, decreasing PF values for increasing number of classes), although the improvement of EV seems to level off for high number of classes, meaning that the gain in stratification skill is only minimal.

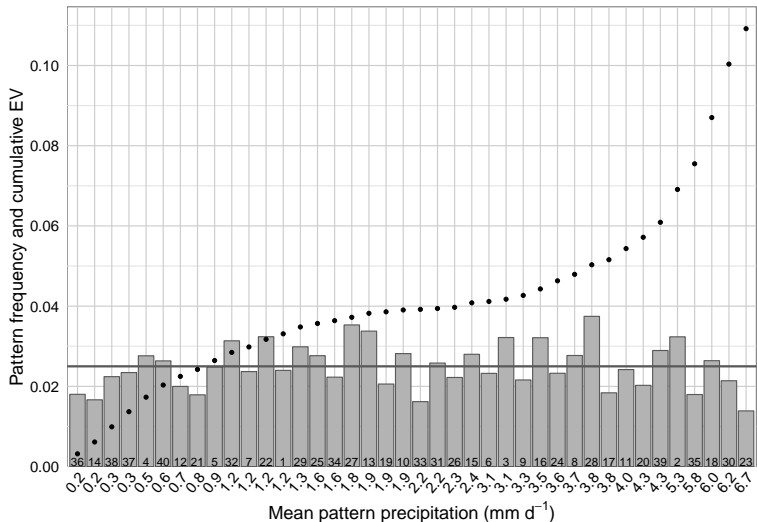

**Figure 7.** Precipitation intensity of patterns in relation to pattern frequency (bars), and cumulated explained variation per pattern (dots). The pattern number is given at the bottom of the bars. Grey horizontal line denotes average frequency to aid distinction of rare and frequent patterns.

### 4.1.3 Stratification skill for precipitation

5 The stratification skill (i.e. EV and PF values) is rather low for precipitation, but maps of mean pattern precipitation (Figure 12) indicate distinct precipitation patterns. Therefore a more detailed investigation of explained variance for individual patterns was done. EV can be expressed as the sum of EV values for individual patterns weighted by the respective relative frequency of the pattern ($n_j/n$):

$$EV = \sum_{j=1}^{k} \frac{n_j}{n} \cdot EV_j \; ; \quad \text{with} \tag{6}$$

$$EV_j = \frac{(\bar{x}_j - \bar{x})^2}{TSS/n} \tag{7}$$

This allows to analyse the contribution of each pattern to the overall EV value. Figure 7 shows the cumulated $EV_j$ values of each pattern. In an idealised case where mean precipitation and frequency of occurrence are uniformly distributed among all types Equation 6 describes (as an integral over a square) a cubic function with a saddle point at the overall mean precipitation. Patterns associated with the tails of the distribution would contribute most to the overall EV, while average types have contributions

15 close to zero (because their mean is close to the overall mean, thus the deviation between both is small, resulting in near-zero $EV_j$).

However, in the case of precipitation, patterns with below-average mean precipitation contribute only little to the overall EV, because the overall mean is rather small (2.4 mm) and hence the deviation between the mean of low-precipitation patterns

and the overall mean is small. This applies to more than half of all patterns (24 out of 40). Most EV contribution is gained by patterns with very high precipitation – 50% of total EV is contributed by the seven patterns with highest precipitation. This behaviour is clearly originating from the strongly right skewed distribution of precipitation. Thus, the small skill values can be considered inherent to precipitation.

Additionally analysing precipitation frequency and intensity per pattern (not shown) reveals that the variations in Figure 12 are mainly caused by pattern-specific precipitation frequency.

## 4.2 Performance of GCMs

After selecting the most appropriate classification, all GCMs (15 models with up to 10 runs for experiment All-Hist) were assigned to the centroids of the final classification, resulting in a catalogue (i.e. time series) of patterns for each GCM run. These time series were compared to the catalogue derived from the reanalysis data to assess the ability of GCMs to reproduce the weather pattern climatology in terms of frequency, seasonality, and persistence as suggested e.g. by Bárdossy et al. (2002). Seasonality is evaluated by the first, last, and peak month of pattern occurrence. All patterns show a distinct seasonality. Each season is characterised by a limited number of consecutive months in which a pattern occurs. We evaluate the beginning (i.e. first month) and end (i.e. last month) of pattern occurrence. The peak month is defined as the month with highest number of days with pattern occurrence. Some patterns show two distinct seasons. In this case both seasons are evaluated separately. Results from different runs of each GCM are averaged.

### 4.2.1 Frequency of patterns

The frequency of patterns as obtained from each GCM run was compared to pattern frequencies in the reanalyses data (Figure 8). The time series are compared for the whole period, i.e. no separation by seasons or individual years was done. Especially for patterns with high mean daily precipitation a good agreement between reanalysis and GCM (All-Hist) would be desirable (maps of average daily values in the Appendix A, Figure 11-16). Frequencies for different runs of one GCM were averaged, but differences between runs are much smaller (usually less than 0.5 %) than between GCMs. The deviations between reanalyses and GCM frequencies are highly diverse for different patterns, e.g. pattern 30 – a high-precipitation pattern with more than 6 mm per day on average (see Appendix A, Figure 12) is well-reproduced, while some GCMs have difficulties to match e.g. patterns 11 or 39. No clear season-specific deviations were found – some models have higher deviations in winter, others in summer (not shown). For eight patterns all GCMs underestimate the frequency found in the reanalysis and for other seven patterns all GCMs overestimate the frequency. By having a closer look into this behaviour, it becomes apparent that particularly cold weather patterns (1, 12, 14, 21, 33, 34, 37) are underestimated, although the warm pattern 27 is also underestimated. Apparently, all GCMs have difficulties in reproducing these weather patterns. However, it goes beyond the scope of this manuscript to analyse the genesis of these weather patterns and why GCMs are not capable to capture them well. With regards to the overestimated patterns (3, 6, 7, 11, 20, 23, 35), they show a tendency towards average to above-average precipitation. But other, high precipitation patterns seem to be well-captured. The remaining 25 patterns enclose the reanalysis values in their range. Among the models with an overall good performance in terms of frequency are CNRM-CM5, GFDL-CM3, and

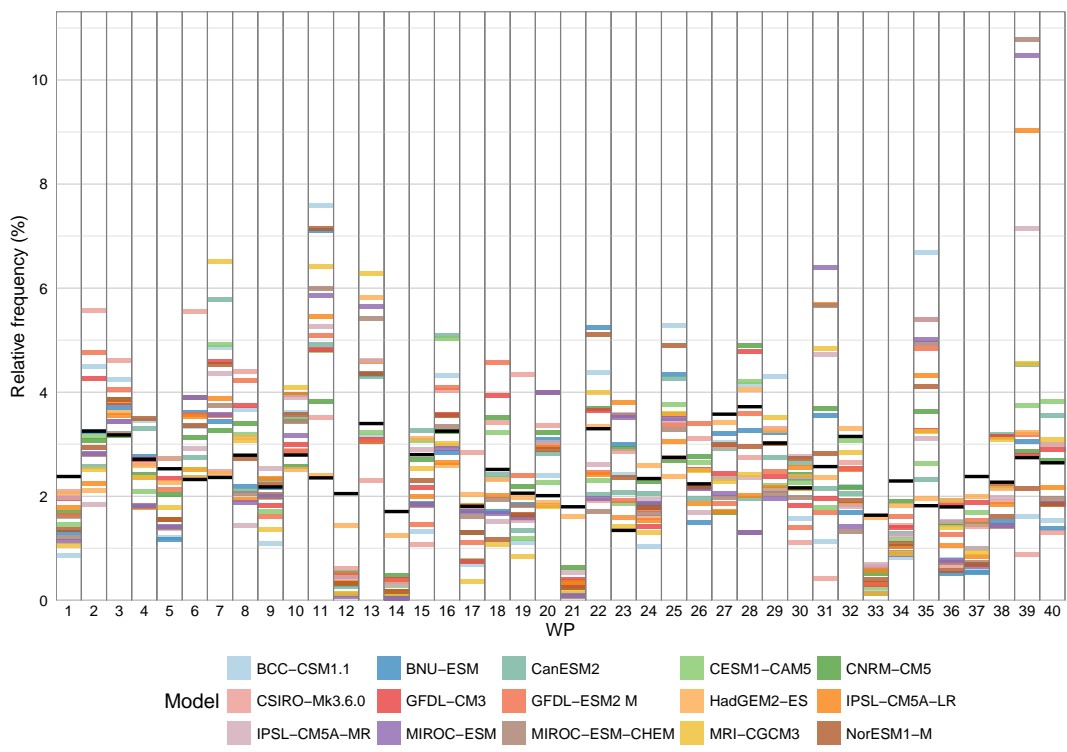

**Figure 8.** Relative frequency of patterns in GCMs (coloured dashes) compared to frequency in reanalysis data (black dashes).

HadGEM2-ES, while the models BCC-CSM1.1, CCSM4, IPSL-CM5A-LR, MIROC-ESM, and MIROC-ESM-CHEM show highest deviations from the reanalyses. In the work of Belleflamme et al. (2014) which uses a similar set of GCMs, three of these bad performing models were found to have best rankings in reproducing pattern frequency (in summer), which shows that statements about GCM performance are somewhat depending on the actual application and its geographic focus.

### 4.2.2 Seasonality

The seasonality of patterns in terms of the earliest and last months of occurrence in the course of the year, and the most frequent month of occurrence is generally well reproduced, even for patterns with two peaks (Figure 9). While start and end are often matched perfectly, the peak months deviate more often, but usually by not more than one or two months. A deviation of one month is considered an acceptably good performance. This good reproduction of pattern seasonality is certainly due to the use of variables with a strong seasonal cycle (temperature and specific humidity) for classification – near-surface temperature and its gradient between continent and sea gives very season-specific patterns that are beneficial for the seasonal stratification of weather patterns.

Most GCMs are able to reproduce the correct start months in 16 to 34 patterns, the highest amount of mismatched patterns (20 or more) are found in BCC-CSM1.1, BNU-ESM, MIROC-ESM, and MIROC-ESM-CHEM. The correct end months are

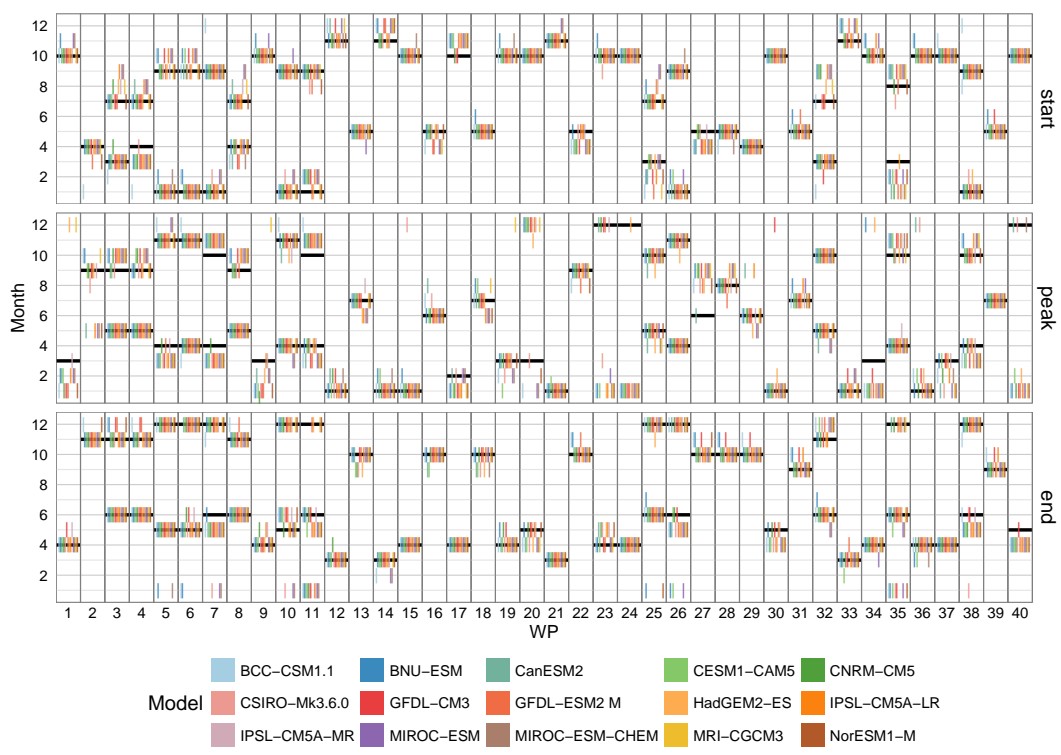

**Figure 9.** Comparison of seasonality of patterns in GCMs (coloured vertical dashes) and reanalysis data (black horizontal dashes). Seasonality is presented as start month(s) (upper panel), peak month(s) (middle panel) and end month(s) (lower panel) of occurrence of patterns. Dashes for GCMs are only vertical to avoid overlapping – each symbol denotes one distinct month. If pattern occurs in two distinct seasons, both are shown.

reproduced in 18 to 32 patterns. Only one GCM with more than 20 mismatched patterns was found (BCC-CSM1.1) and 15 or more mismatches occurred in BNU-ESM and CESM1-CAM5. Models BCC-CSM1.1, BNU-ESM, IPSL-CM5A-LR, MIROC-ESM, MIROC-ESM-CHEM, and MRI-CGCM3 fail in more than half of all patterns to match the peak months. All GCMs are
30   generally slightly better in capturing the correct start and end month of summer or winter patterns compared to spring/autumn patterns.

### 4.2.3   Persistence

Finally the persistence of patterns is assessed as the number of consecutive days with the same weather pattern. In Figure 10 the average duration in reanalysis data is compared to the duration in GCMs. The mean duration of patterns is mainly around 2 days, which is usually well represented by the GCMs. Deviations from the persistence of reanalysis data that are greater than
1 day were only found in very few patterns (14, 39), usually mean persistence deviates by less than 1 day.

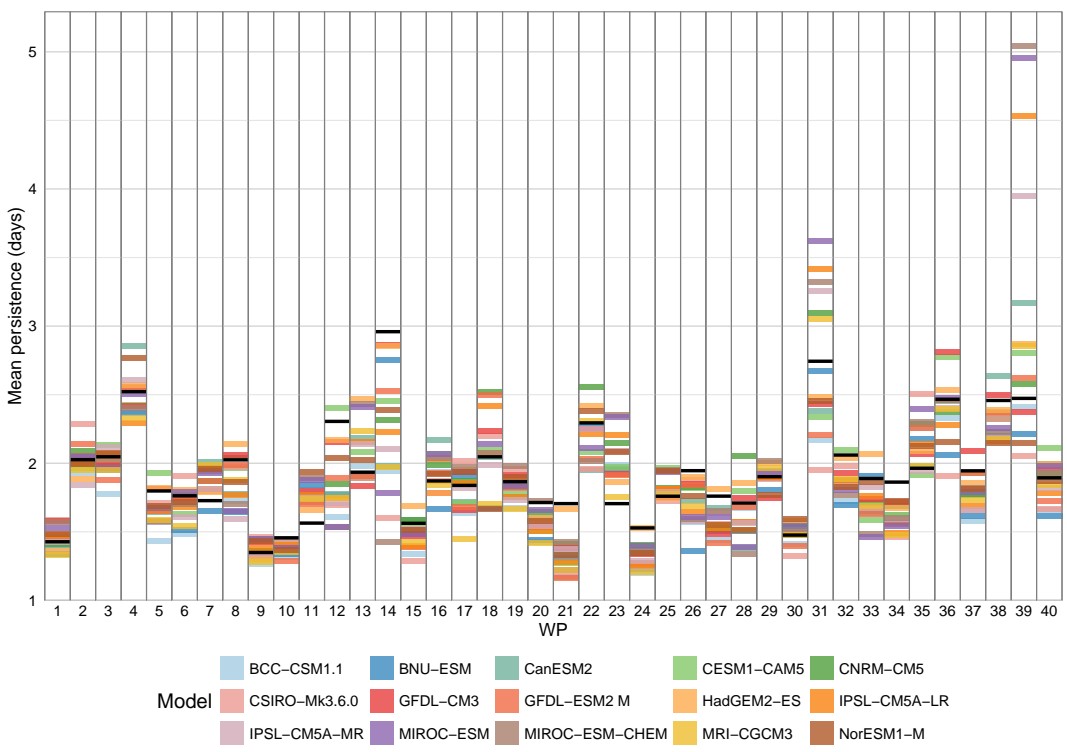

**Figure 10.** Persistence of patterns (mean number of consecutive days with same pattern) in GCMs (coloured dashes) compared to persistence on reanalysis data (black dashes).

Best agreement between reanalyses and GCMs was found for CESM1-CAM5, CNRM-CM5, GFDL-CM3, and HadGEM2-ES, while greatest deviations occurred for BCC-CSM1.1, CSIRO-Mk3.6.0, IPSL-CM5A-LR, MIROC-ESM, and MIROC-ESM-CHEM. There is no general difference in deviation from reanalysis for different seasons, though most GCMs are matching persistence of spring/autumn patterns slightly better than persistence of summer or winter patterns. Other studies found patterns to last longer than in our case (e.g. Kyselý, 2007, who found mean persistence for Hess-Brezowsky-Grosswetterlagen of 4.3–5.2 (and up to 6.2) days), which might be due to our comparatively large number of patterns.

## 5 Discussion

### 5.1 On the optimal classification

This study derives an "optimal" weather pattern classification for the Rhine catchment and investigates to which extent weather patterns are able to stratify local climate variables. Furthermore, the ability of the latest GCM generation to reproduce these weather patterns is evaluated in terms of frequency, seasonality and persistence. The particularities of this study, compared to

past studies on weather pattern classifications, include (1) the investigation of the skill of several classification variables, (2) the large number of local weather variables used for classification evaluation, (3) the large study area $(160\,000\,\mathrm{km}^2)$ and the very high number of climate stations (490), and (4) the use of long time series (111 years).

It has been argued that there is no "best" classification and that the optimal solution depends on the specific application and region. The best classification for the Rhine catchment was achieved with a combination of mean sea level pressure, temperature and specific humidity as classification variables. Often, weather patterns are classified on pressure fields only. Our results suggest that adding humidity and temperature, which exhibits a distinct seasonal cycle, as classification variable improves the stratification of local climate variables considerably and support the findings of Goodess and Jones (2002). Including temperature as classification variable, yields a very good stratification of weather patterns throughout the year, i.e. weather patterns also show a distinct seasonality. In this way a single classification can be used for the whole year, and there is no need to provide different classifications for each season separately contrary to classifications based solely on mean sea level pressure.

Concerning the number of classes, our results do not give a clear indication about the optimal number. We have selected a comparatively large number, i.e. 40 patterns. This selection is in line with other studies that compared classifications. Philipp (2009) found for SANDRA classifications that best skills are reached for class numbers greater than 30. Tveito (2010) compared 73 classifications from the COST733 collection of classifications catalogues and found best performances for high numbers of classes; generally for the same classification method a solution with more types performed better. The ten best classifications had at least 26 classes and the best three classifications had 30, 40, and 29 types, respectively. Of course, the decision about the number of classes is guided by the purpose of the classification and the data availability. The stratification of local climate variables into a large number of classes requires sufficient amount of data. Our sensitivity analysis with 100 weather patterns clearly indicated worse performance compared to the classification based on 40 patterns. But in general, a larger number of classes is advisable if not limited by the amount of available data.

In terms of spatial domain, the best results are obtained for rather small classification areas covering the target area. Increasing the classification domain covering the whole of Europe slightly aggravates the stratification of local variables, particularly of temperature. It is however, difficult to draw generalisations with regards to the selection of the spatial domain given our results.

The "optimal" classification is only partially able to stratify local climate variable, i.e. the classification explains a modest share of the local climate variability. EV values, averaged across all 490 stations in the Rhine catchment, are in the range of 10–20% for precipitation, 70–80% for temperature, 10–20% for humidity and 40% for radiation. Hence, especially local precipitation and humidity are governed by processes that are not completely represented by the large-scale distribution of pressure, temperature and humidity. This result questions the widespread downscaling approaches that are based on weather pattern classification. The within-type variability dominates versus the between-type variability, at least for local precipitation and humidity. Before applying the weather pattern based downscaling approach, it should therefore be investigated whether the link between the large-scale synoptic situation and the local climate variable of interest is strong enough for the given purpose.

Although downscaling approaches based on weather patterns are widespread, there are not many studies that have assessed the skill of weather patterns for stratifying local climate variables. The available studies report skill values that are comparable to our results. For example, Osborn and Jones (2000) found large residuals between precipitation predicted from circulation indices and observed precipitation. Enke and Spekat (1997) obtained 20.5% of explained variation for precipitation and 80.9% for mean temperature. Huth et al. (2016) compared a large number of classifications from COST ACTION 733 using different classification methods, numbers of patterns, spatial domains, classification variables, sequence lengths of 1 or 4 days. For all domains and classification settings they obtained EV values of max. 0.33 for precipitation and max. 0.46 for mean temperature. The much higher values for temperature in our study can be explained by the use of 2 m temperature as additional classification variable. Our classification using only sea level pressure obtains similarly low values. For those classifications that are best comparable to our study, i.e. method SANDRA, whole year, 1-day sequence, classification on sea level pressure, 9, 18, or 27 types, comparable spatial domain, they obtain EV values of 0.07–0.28 for temperature and 0.08–0.27 for precipitation. These results are averages across all seasons, whereas the results for the winter are generally better. The study of Enke et al. (2005b) suggested that classifications that are highly optimised towards a certain local climate variable, such as precipitation, may have significantly better skill than classifications for several variables. However, highly optimised classifications have the disadvantage that their skill deteriorates when applied for other target variables.

Downscaling using the weather pattern approach is based on the assumption of a time-constant relationship between patterns and local climate variables. Instationarities in the relationship between weather types and local variables is a long-debated issue in downscaling (IPCC, 2007), and several studies indicated their presence (e.g. Widmann and Schär, 1997; Beck et al., 2007; Haberlandt et al., 2015). Those classifications were, however, based on sea level pressure only (Beck et al., 2007; Haberlandt et al., 2015) or additionally included geopotential height (Widmann and Schär, 1997). The addition of temperature and specific humidity might provide a better classification also in terms of capturing transient changes in local climate by changes in weather pattern sequencing. This suggestion is supported by the regional climate simulations of Schär et al. (1996). For the European Alps, they found that increased warming can lead to larger moisture fluxes and larger precipitation rates even when the synoptic situation remains unchanged. Thus, it should be further investigated whether classifications that are based on additional variables besides pressure fields show less instationarity in the link between synoptic situation and local climate.

## 5.2  On the skill of GCMs

Concerning the skill of the latest generation of GCMs to reproduce these weather patterns, we find that the main characteristics of weather patterns derived from ERA20C reanalysis data are well represented in GCMs that are forced with observed GHG concentrations. Interestingly, the performance of GCMs is usually similar for a certain GCM for the analysed characteristics, i.e. frequency, seasonality, or persistence of patterns. This result suggests that some GCMs are much better suited for downscaling based on weather pattern classifications. Others should be excluded or their results should at least be interpreted with greatest care. From the results obtained, it would be advisable not to consider the models BCC-CSM1.1, MIROC-ESM, and MIROC-ESM-CHEM. This would leave 12 GCMs with acceptable performance. However, it should be noted that the skill of GCMs

may depend on the specific classification, i.e. the classification variables and the region. Another classifications might result in a different ranking of GCMs.

## 6 Conclusions

In the scope of an attribution study aimed at quantifying the role of climate change, in particular the contribution of anthropogenic climate change, to changes in flood flows in the Rhine catchment, we developed a weather pattern classification. This classification is intended to be used for downscaling of general circulation model outputs with a multi-site, multi-variate weather generator. An optimal classification was selected by evaluating four different combinations of classification variables based on the ERA20C reanalysis data, by testing six spatial domains and four numbers of classes. The best stratification of local variables (daily precipitation, humidity, radiation, and mean, minimum, and maximum temperature) was obtained when using 40 classes from the SANDRA classification, with sea level pressure, temperature and specific humidity combined over a relatively small Central European domain. The performance of different classifications was assessed with Explained Variation (EV) and Pseudo-F statistic. The optimal classification showed rather high EV (similar to Pseudo-F statistic) for single variables except precipitation and humidity. A multi-variate evaluation demonstrates that the classification is reasonable, although single variables are not very well stratified. Different weather patterns can be similar in one variable (e.g. temperature), but exhibit very distinct behaviour in others (e.g. precipitation). Often, weather patterns are classified on pressure fields only. Our results suggest that adding humidity and temperature as classification variables improves the stratification considerably. This results in a very good stratification of weather patterns throughout the year. In this way a single classification can be used for the whole year, and there is no need to provide different classifications for each season. Adding further classification variables to pressure fields may also alleviate the often encountered problem that the link between the synoptic situation and the local climate is not constant in time.

GCMs should properly reproduce the climatology of weather patterns in order to be applicable for the attribution of flood changes. Hence, the performance of 15 GCMs from the CMIP5 project in matching the climatology of ERA20C reanalysis in terms of frequency, seasonality (month of occurrence) and persistence (number of consecutive days) of weather patterns was evaluated. The frequency of weather patterns is matched well by the majority of GCMs with a few GCMs showing systematic deviations. No season-specific deviations were found. Due to the use of temperature for pattern classification, the seasonality of weather patterns matched well in most of the GCMs. All GCMs were found able to better capture the seasonality of summer and winter patterns compared to spring and autumn ones. The mean duration of patterns was about 2 days with most GCMs being able to reproduce this persistence. Overall, three GCMs BCC-CSM1.1, MIROC-ESM, and MIROC-ESM-CHEM were found to systematically deviate from the reanalysis weather pattern climatology. The variation between different realisations of one GCM was found small compared to the difference between various GCMs.

**Appendix A: Maps of meteorological mean values for each pattern**

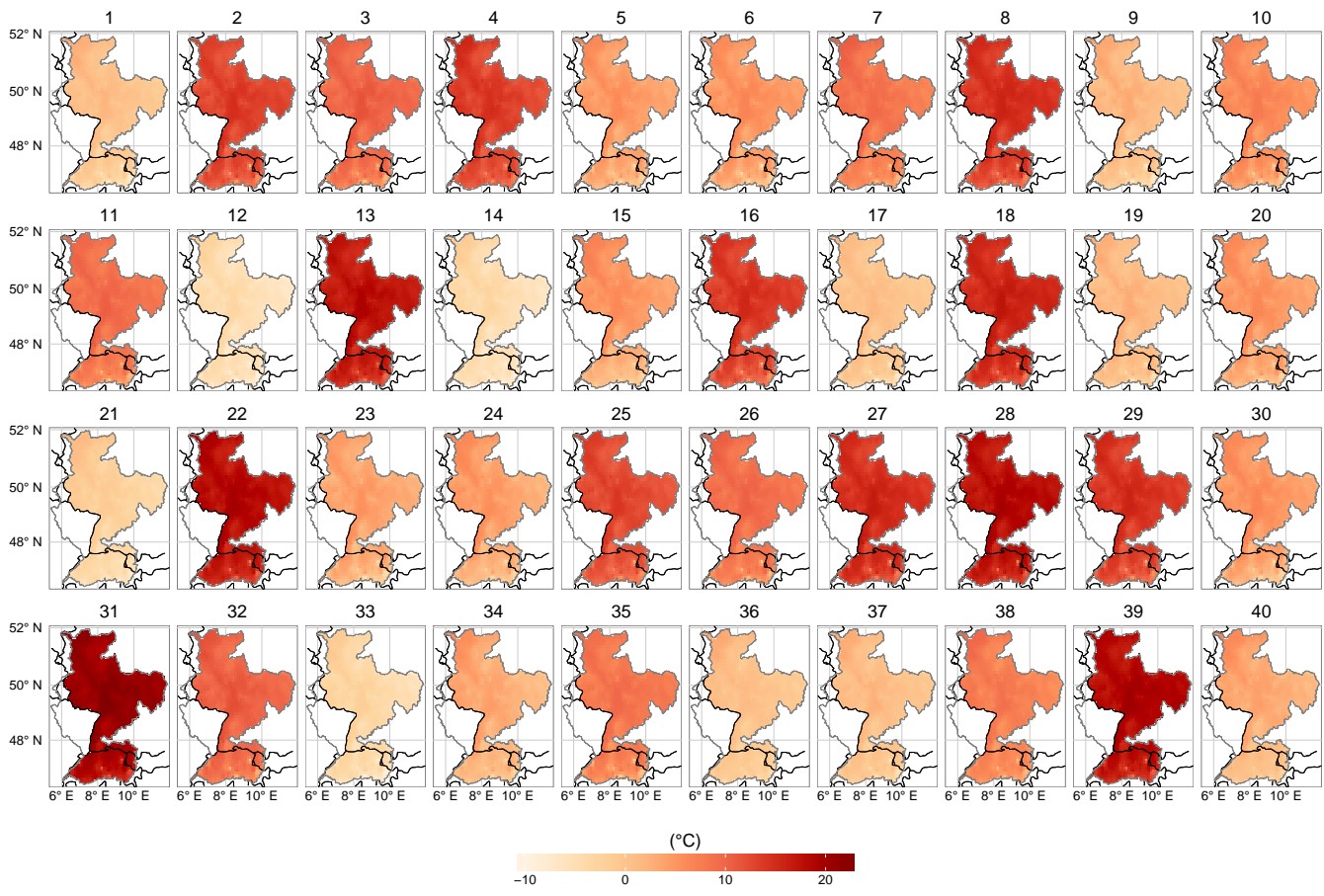

**Figure 11.** Average (over all days with the respective pattern) mean temperature for all weather patterns. Black lines denote state borders, grey line Rhine catchment.

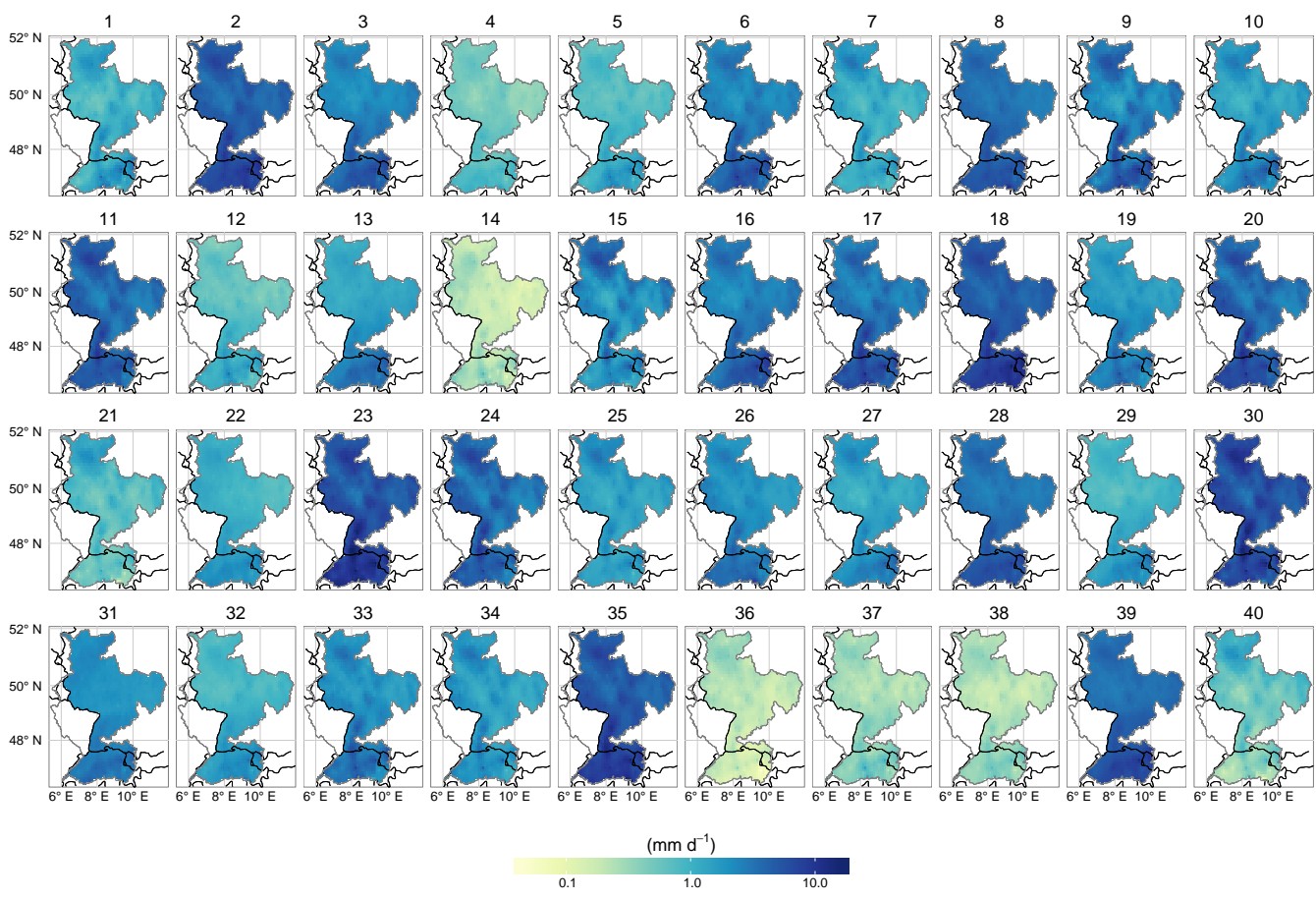

**Figure 12.** As in Figure 11, but for daily precipitation.

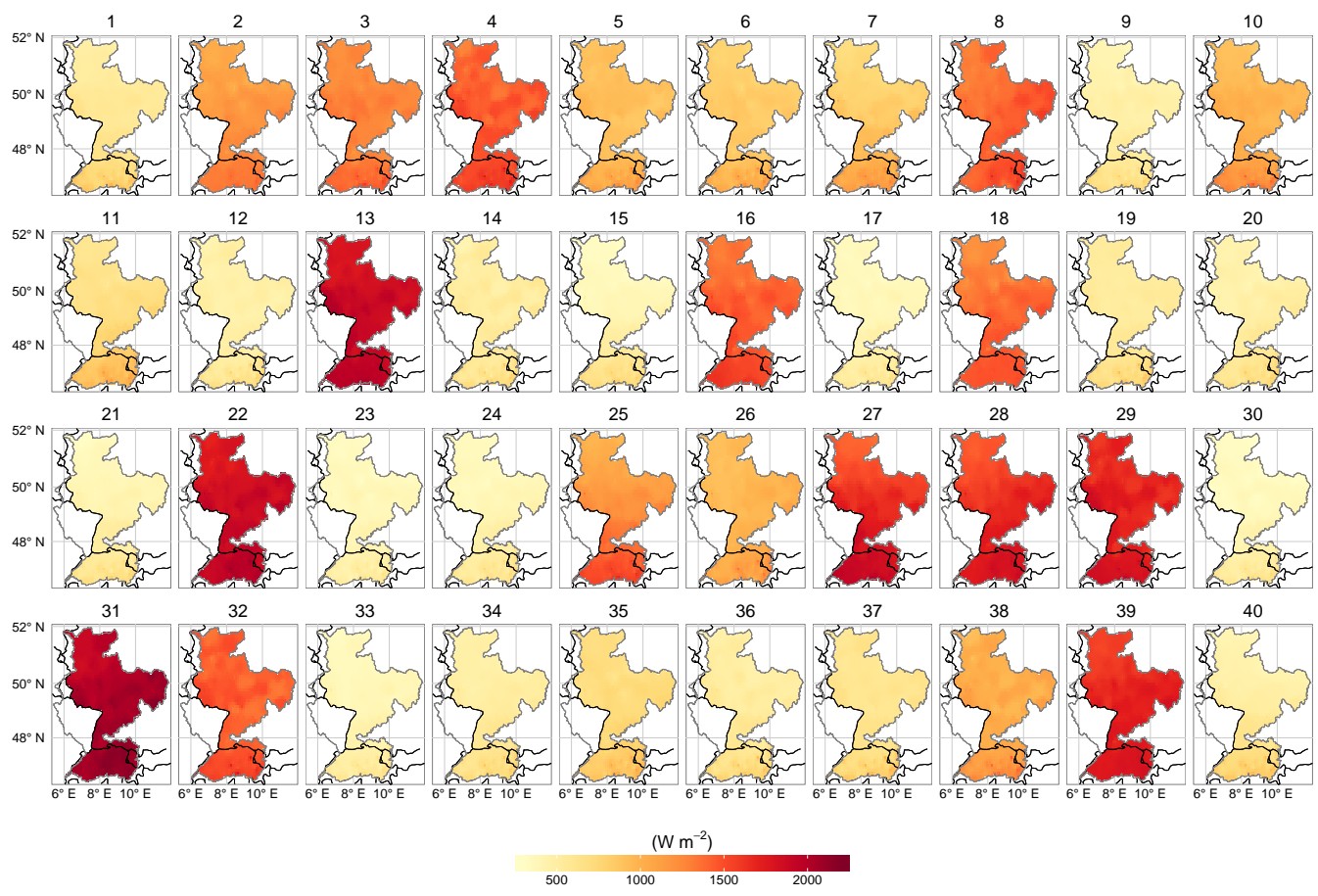

**Figure 13.** As in Figure 11, but for global radiation.

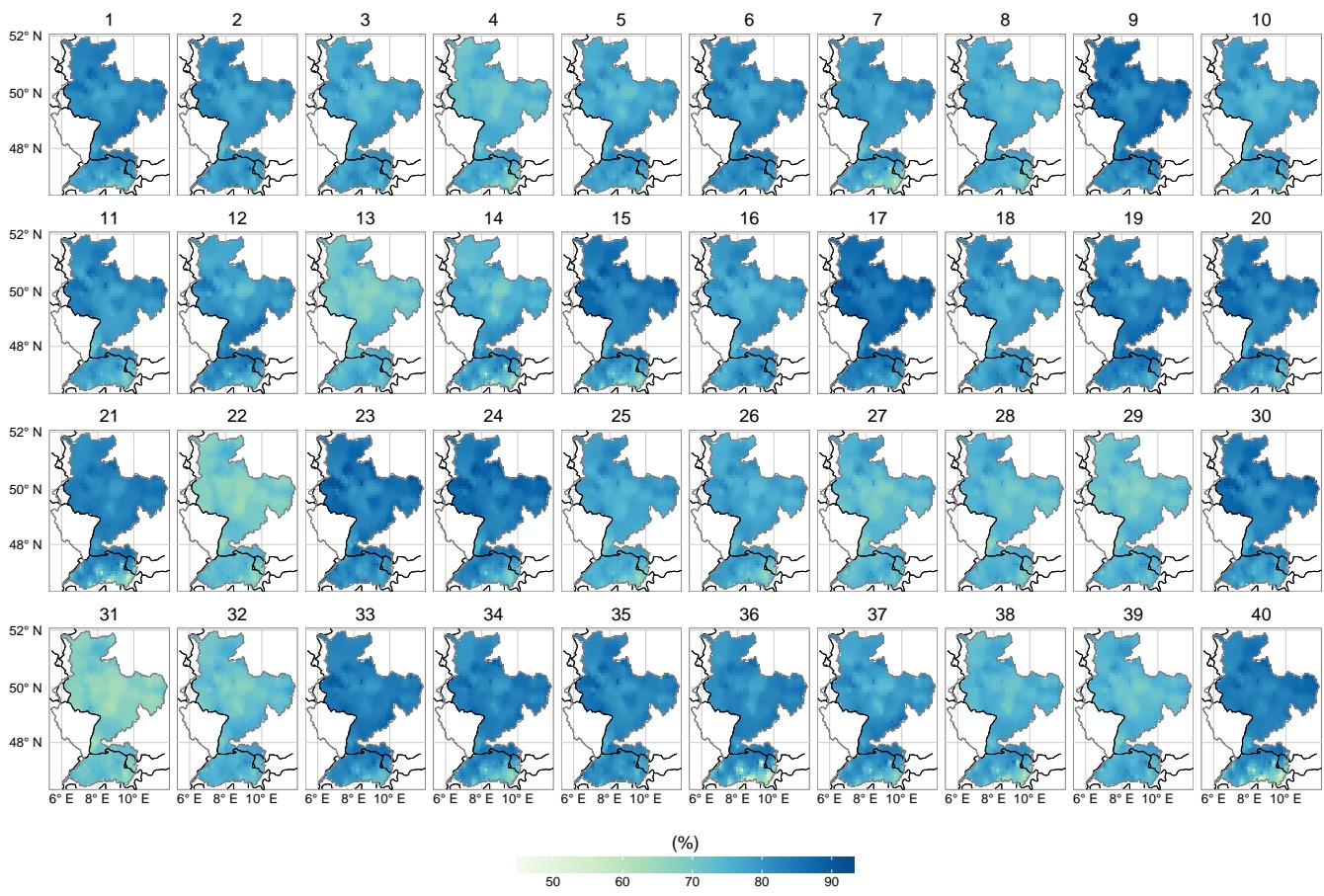

**Figure 14.** As in Figure 11, but for relative humidity.

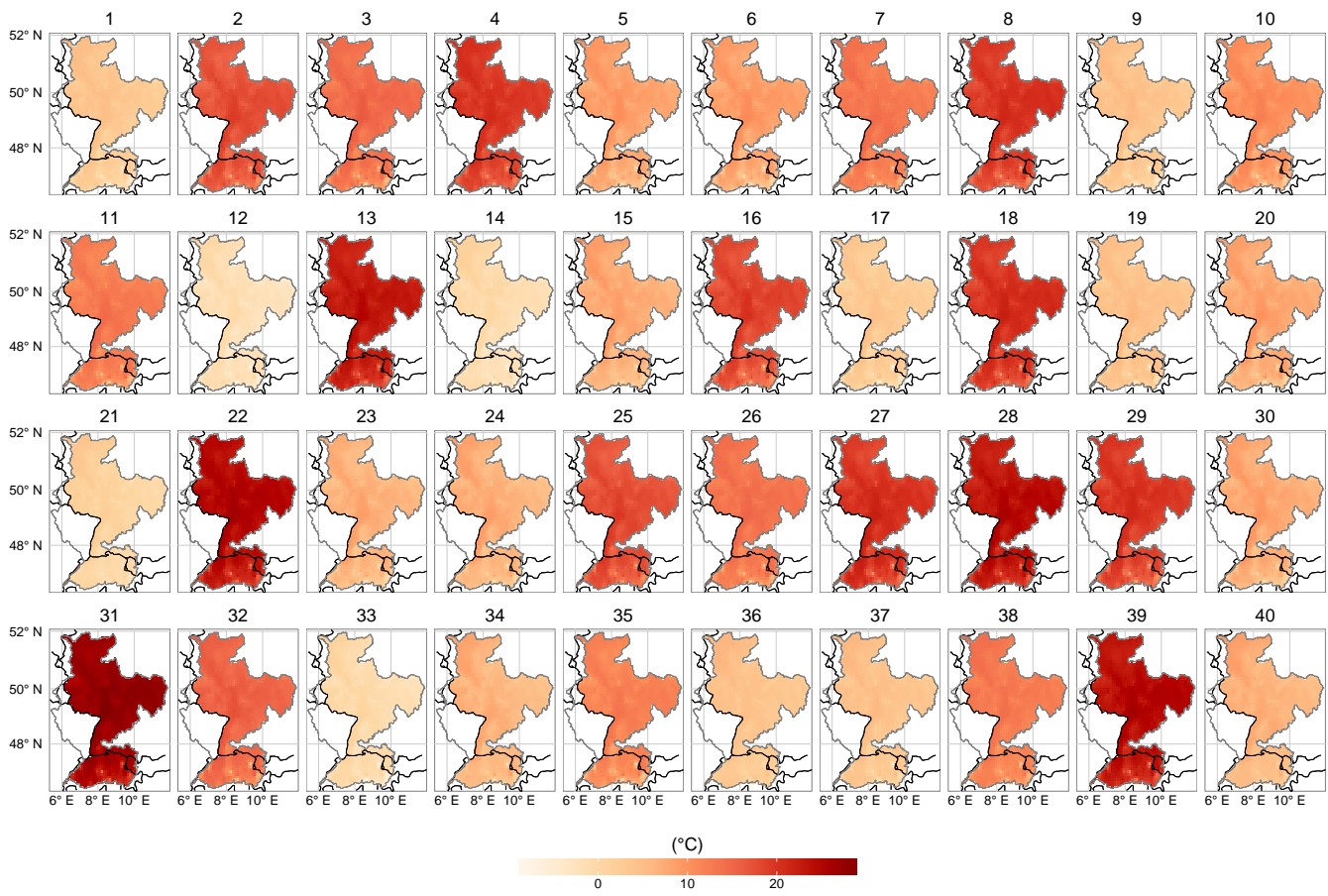

**Figure 15.** As in Figure 11, but for daily maximum temperature.

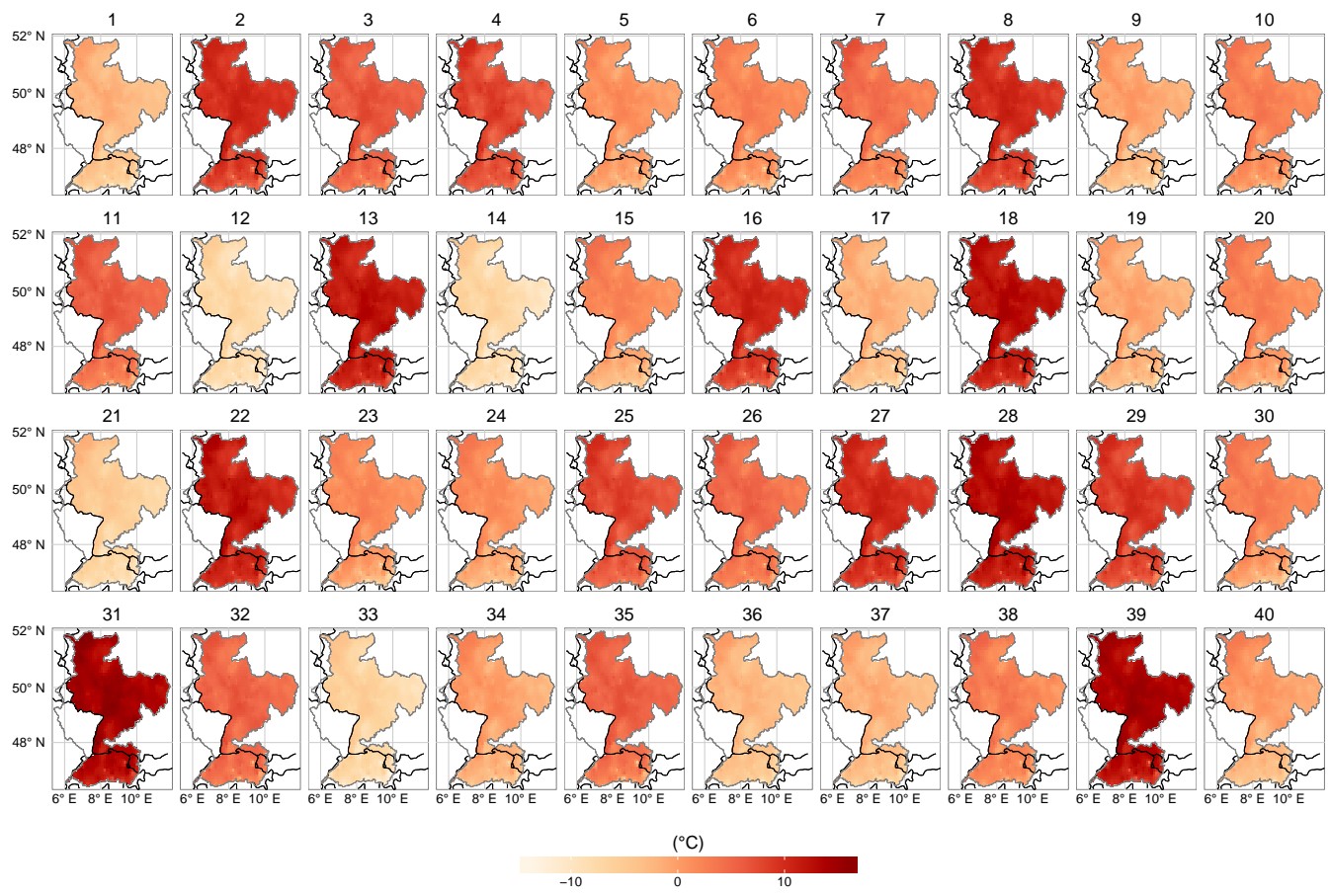

**Figure 16.** As in Figure 11, but for daily minimum temperature.

*Author contributions.* All authors contributed to the experiment design and the manuscript. A. Murawski performed the computations and created the figures.

*Acknowledgements.* We acknowledge the World Climate Research Programme's Working Group on Coupled Modelling, which is responsible for CMIP, and we thank the climate modeling groups (listed in Table 1 of this paper) for producing and making available their model output. For CMIP the U.S. Department of Energy's Program for Climate Model Diagnosis and Intercomparison provides coordinating support and led development of software infrastructure in partnership with the Global Organization for Earth System Science Portals. (from http://cmip-pcmdi.llnl.gov/cmip5/citation.html)

The ECMWF ERA-20C Reanalysis data used in this study were obtained from the ECMWF Data Server (http://www.ecmwf.int).

We gratefully appreciate the provision of data by the national meteorological services of Germany, Austria, and Switzerland, kindly provided and processed by the Potsdam-Institute for Climate Impact Research (PIK).

Help and discussion on the classification software *cost733class* by Dr. Tobias Pardowitz (FU Berlin) is greatly appreciated.

A. Murawski acknowledges funding by Climate KIC.

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

**Table 1.** Overview on GCMs used (http://esgf-data.dkrz.de/).

| Model | Institute ID | Country | Period | Resolution Lon × Lat | Runs |
|---|---|---|---|---|---|
| BCC-CSM1.1 | BCC | China | 1850 – 2012 | 2.8 × 2.8 | 3 |
| BNU-ESM | GCESS | China | 1950 – 2005 | 2.8 × 2.8 | 1 |
| CanESM2 | CCCMA | Canada | 1850 – 2005 | 2.8 × 2.8 | 5 |
| CESM1-CAM5 | NSF-DOE-NCAR | USA | 1850 – 2005 | 1.2 × 0.9 | 1 |
| CNRM-CM5 | CNRM-CERFACS | France | 1850 – 2005 | 1.4 × 1.4 | 10 |
| CSIRO-Mk3.6.0 | CSIRO-QCCCE | Australia | 1850 – 2005 | 1.9 × 1.9 | 10 |
| GFDL-CM3 | NOAA-GFDL | USA | 1860 – 2005 | 2.5 × 2.0 | 3 |
| GFDL-ESM2 M | NOAA-GFDL | USA | 1861 – 2005 | 2.5 × 2.0 | 1 |
| HadGEM2-ES | MOHC | UK | 1859 – 2005 | 1.9 × 1.2 | 4 |
| IPSL-CM5A-LR | IPSL | France | 1850 – 2005 | 3.8 × 1.9 | 6 |
| IPSL-CM5A-MR | IPSL | France | 1850 – 2005 | 2.5 × 1.3 | 3 |
| MIROC-ESM | MIROC | Japan | 1850 – 2005 | 2.8 × 2.8 | 3 |
| MIROC-ESM-CHEM | MIROC | Japan | 1850 – 2005 | 2.8 × 2.8 | 1 |
| MRI-CGCM3 | MRI | Japan | 1850 – 2005 | 1.1 × 1.1 | 5 |
| NorESM1-M | NCC | Norway | 1850 – 2005 | 2.5 × 1.9 | 3 |