# Peer review of "Can local climate variability be explained by weather patterns? A multi-station evaluation for the Rhine basin"

_Hydrology and Earth System Sciences, 2016_

## Referee Comment (RC1) · Anonymous Referee #1 · 18 Jul 2016

Notes concerning

Can local climate variability be explained by weather patterns? A multi-station evaluation for the Rhine basin

Aline Murawski, Gerd Bürger, Sergiy Vorogushyn and Bruno Merz

Hess-2016-286

There was an initial reaction upon studying the manuscript: 40 patterns?? Are you serious??

Then another question arose: Are pressure fields as classification a good basis for the stratification of data? There is good experience with using relative topography instead (Spekat et al., 2010 – reference is given at the end).

The authors do not go into seasonality when analyzing their data. There is contrary experience from the COST733 Action on classification methods (in the meantime, the results of that COST Action have been made publicly available, see reference at the end). Plus there is experience towards great usefulness of seasonality in classification, indicated, e.g., in Spekat et al. (2010).

Usage of E-OBS data – Fig. 1 only shows 8 E-OBS grid points. Text mentions 10. Moreover in Fig. 1: Poor choice of colours for dots.

From page 5 on the line numbering is irritatingly confusing. Just look at the repetition of line number 5 on page 5…

Section 3.2 page 7 line 12: Maybe relative topography is not directly available, but contributing geopotentials can be easily extracted and the retop could be easily computed.

Page 7 line 14 – and I mean the second appearance of this line (sigh…) beginngin with "(extents " it must be made more clear that these numbers refer to geographic degrees of latitude and longitude.

Page 10 figure 3: Not clear what the numbers "12" , "14" and "33" on the right-hand side of the array of figures mean. I would furthermore recommend to deliberately use different colour schemes for different parameters, so they can be better distinguished. Even if the authors would not follow this suggestion at least they should reverse the assignment of the colours (left side red, right side blue) for PREC and HUMID since blue would then point to wet condition which suits the intuition better. More a thought than a substantial comment: All selected patterns are rather cold/have rather low radiation, so perhaps one would like to see examples for classes which denote different conditions.

Page 10 line 8 I have doubts if retaining the absolute values is a good approach when you have season-specific classes. Particularly temperature is way different from season to season. So *au contraire* to what the authors write, anomalies are a good way forward because they cover a similar value range in all seasons. Moreover, there is the experience that a further reduction in the number of classes is possible by using anomalies (this would be favourable in the light of the big set of 40 classes used by the authors).

Page 11 figure 4: I suggest to add y-axis labels on the right-hand side, too. Further suggestion: Use open triangle and open circles which are better visible in case of overlaps, and those do appear frequently.

Page 12 figure 5: It would be could to have the results for four different class number shown at least for one more extent (or domain size, as one might better call it).

Page 12 line 14: "increasing EV" - this is very minute if you look at it. Therefore I suggest to write "almost no change".

Page 14 figure 7: I am amazed how relatively even the frequency distribution is. Expectation would be that some classes would be quite rare. Furthermore: is the property displayed really the quotient of frequency and cumulative EV? Furtherfurthermore, what does the (-) at the end of the y axis label mean?

Page 14 lines 30 and following as well as Fig. 8: It is remarkable and should be pointed out that for numerous classes the reanalyses (black dashes) mark either the lowest or the highest frequency so in those cases ALL GCMs are unanimously indicating either higher or lower frequencies, respectively. Isn't that an odd behavior?

Page 14/15 Section 4.2.2 I assume that Fig. 9 on Page 16 is meant to visualize this, right? Then make a reference to that figure!

Section 4.2.2 again: A definition needed is needed as to what is considered a *good* reproduction. Imagine that there could be ties in the months of most frequent occurrences - or months with very similar frequency. Would that still be good/superior/inferior reproductions, then?

Page 15 Section 4.2.3 The text points to Figure 8, wheras the reference should point to Figure 10.

Page 18 around line 35, the aspect of stratification skill was presented in Spekat et al. (2010), too. Perhaps this needs to be mentioned in the text.

Page 18 line 12 (bottom): More like a comment - this discussion opens up a whole philosophical can of worms, i.e., universality versus optimization. Should the goal be to find a classification able to cover "everything but with a variable degree of fidelity" or should the goal be to find a classification that is region- and variable-specific, yet has a high skill? Maybe the authors could be drawn into discussing this for a bit, too?

General comment on the "Discussion" section: It is rather long (no criticism concerning the length, mind you) and could benefit from the insertion of subsections.

Page 34 Table 1: Is it "Runs" or "Run", i.e., did the authors use all 10 CNRM runs or all 10 CSIRO runs (for example) or did they use just a specific one of those? Then this particular run should be specified. This refers back to page 3 line 12 where it is ambiguous if ALL or SOME runs were analyzed in this paper.

General comment concerning Figs. 11 thru 16: It is amazing to me that those 40 classes, some of which are *visually* quite similar to each other, apparently constitute sets of necessary distinctiveness. Just from looking at them the, admittedly subjective, estimate would be that much fewer classes should be sufficient.

General comment on the line numbering: It should be uninterrupted, starting with 1 and end in the high several hundreds. The numbering in the draft here is misguided and misguiding.

General comment on the figure placement: Particularly for Figs. 7 thru 10, a better proximity to their mentioning in the text and respective paragraphs to which they belong should be found.

General comment on Literature – one could of course think of a "me too" effect… - but there is a paper from 2010 which covers or complements several aspects of the manuscripts's reasoning:
Spekat, A., F. Kreienkamp and W. Enke, 2010: An impact-oriented classification method for atmospheric patterns. – Physics and Chemistry of the Earth **35**, 352-359.
Also, perhaps unknown to the authors of the manuscript, the final report for the COST733 Action is now available. The link to the final report is: https://opus.bibliothek.uni-augsburg.de/opus4/frontdoor/index/index/docId/3768

That link is permanent. The URN is urn:nbn:de:bvb:384-opus4-37682 (it can be found using a web search engine).

So, bottom line: Something in between minor and major revision. Some reasoning needs to be better shaped, some needs to mention a bit more what alternative paths have been pursued. There is some potential to improve technical aspects (figures, mostly) and general understandability.

---

## Author Comment (AC1) · 27 Jul 2016

**Reviewer #1:**

We thank the reviewer for his/her constructive comments on our manuscript and provide point-by-point response hereafter.

There was an initial reaction upon studying the manuscript: 40 patterns?? Are you serious??

Yes, we are. Number 40 for the number of classes is not unusual (Philipp et al., 2009). We give various reasons in the discussion (p 17f) and in chapter 3.2 for choosing this number of classes. Among others, we need high stratification of variables for a multi-variate weather generator, which will be subsequently used for global model downscaling. This is best apporoached by a high number of classes. The availability of a long daily time series of daily climate observations (111 years) in the Rhine catchment further justifies our selection.

Then another question arose: Are pressure fields as classification a good basis for the stratification of data? There is good experience with using relative topography instead (Spekat et al., 2010 – reference is given at the end).

No doubt, there are other suitable variables to be used for a classification besides pressure fields. By the way, our 'optimal' classification is based on a combination of sea level pressure, temperature and humidity.  As we stated in chapter 3.2: "However our selection was restricted to variables that are also available from the GCM outputs." Since the classification is to be used for an attribution study later on, we need to select variables for classification that are available for "historical" and "historicalNat" runs of the CMIP5 project. To date geopotential height is only available for 3 Models (10 runs in total) of the historicalNat experiment and 9 Models of the historical experiment. Thus, we would reduce our data base considerably if we decided to use geopotential height or any parameter derived from that. Geopotential height is related to temperature and pressure. We show in Fig. 6 that a classification that uses geopotential height (among other variables) does not perform better than our 'optimal' classification, which includes mean sea level pressure and temperature.

The authors do not go into seasonality when analyzing their data. There is contrary experience from the COST733 Action on classification methods (in the meantime, the results of that COST Action have been made publicly available, see reference at the end). Plus there is experience towards great usefulness of seasonality in classification, indicated, e.g., in Spekat et al. (2010).

We do mention (and evaluate) the seasonality of our patterns (chapter 4.1.1 p 10). We show a distinct pattern seasonality due to the use of temperature as classification variable. By using absolute temperature (not anomalies) and a relatively large number of patterns, we achieve seasonal pattern stratification. We do not establish one classification per season, i.e. we "allow" our patterns to assign to their respective season by themselves (and to move within the year under the effect of climate change, if necessary). To make it short: we account for seasonality implicitly (in contrast to other studies that use one separate classification per pre-defined season).

Thanks for pointing out to the COST733 Action final report. Many (most) results of the report have been already published some years ago and we are well aware of them (many being cited in the manuscript).

Usage of E-OBS data – Fig. 1 only shows 8 E-OBS grid points. Text mentions 10. Moreover in Fig. 1: Poor choice of colours for dots.

The figure has been adapted, thanks for pointing this out. Regarding the 10 E-OBS points: it is 10, just two of them are very close to the dark red line denoting the Rhine catchment – probably they have been overseen with the previous set of colours.

From page 5 on the line numbering is irritatingly confusing. Just look at the repetition of line number 5 on page 5…

True… the manuscript has been compiled using HESS' Latex template. Apparently something went wrong. It is fixed now.

Section 3.2 page 7 line 12: Maybe relative topography is not directly available, but contributing geopotentials can be easily extracted and the retop could be easily computed.

See answer to the second comment.

Page 7 line 14 – and I mean the second appearance of this line (sigh…) beginngin with "(extents " it must be made more clear that these numbers refer to geographic degrees of latitude and longitude.

This has been adapted now in the text (last paragraph of 3.2) and in the caption of figure 2.

Page 10 figure 3: Not clear what the numbers "12" , "14" and "33" on the right-hand side of the array of figures mean. I would furthermore recommend to deliberately use different colour schemes for different parameters, so they can be better distinguished. Even if the authors would not follow this suggestion at least they should reverse the assignment of the colours (left side red, right side blue) for PREC and HUMID since blue would then point to wet condition which suits the intuition better. More a thought than a substantial comment: All selected patterns are rather cold/have rather low radiation, so perhaps one would like to see examples for classes which denote different conditions.

The numbers 12, 14, and 33 indicate the pattern number that have been selected. This has been pointed out now in the text (second last paragraph of 3.3) and in the caption of figure 3.

The colour scales have been adapted.

Regarding the comment concerning pattern selection: the point of this selection is to show patterns that are very similar in some variables (in this case temperature and radiation), but differ in other variables (here: precipitation and humidity). This aims at pointing out the need of multi-variate classification as we describe in the second last paragraph of 3.3.

Page 10 line 8 I have doubts if retaining the absolute values is a good approach when you have season-specific classes. Particularly temperature is way different from season to season. So *au contraire* to what the authors write, anomalies are a good way forward because they cover a similar value range in all seasons. Moreover, there is the experience that a further reduction in the number of classes is possible by using anomalies (this would be favourable in the light of the big set of 40 classes used by the authors).

We would like to refer the reviewer to our answer to his/her third comment. We do not define the patterns explicitly for each season. This is basically achieved through using the absolute values for temperature as classification variable. The advantage is that we have a continuous sequence of patterns. The fact that each pattern occurs only in a rather limited number of months is solely due to the usage of absolute temperature values for classification. We do know that anomalies are used in many other papers and we are well aware that the annual signal of temperature dominates our classification. See e.g. second paragraph of 4.1.1. Nevertheless, regarding the future use of that classification (downscaling tool in climate change attribution), we appreciate the seasonal restriction of patterns.

Page 11 figure 4: I suggest to add y-axis labels on the right-hand side, too. Further suggestion: Use open triangle and open circles which are better visible in case of overlaps, and those do appear frequently.

The figure have been updated using open symbols now. However, we would like to refrain from adding a second y axis. The grid lines in the plots should be sufficient to guide the reader's eye.

Page 12 figure 5: It would be could to have the results for four different class number shown at least for one more extent (or domain size, as one might better call it).

We would prefer to refrain from that as well. The parameter selection follows a clear strategy, as we point out in the last paragraph of 3.2: 1. Select variable, 2. Select spatial domain, 3. Select number of classes. Adding further information into the figures would jeopardize its clarity.

Page 12 line 14: "increasing EV" - this is very minute if you look at it. Therefore I suggest to write "almost no change".

The mentioned paragraph states: "(…) confirming the general tendency (increasing EV, decreasing PF values for increasing number of classes), although the improvement of EV seems to level off for high number of classes, meaning that the gain in stratification skill is only minimal." – Increasing tendency refers to the fact that a higher number of classes results in higher EV values (which is evident from figure 5). Of course, as the reviewer has mentioned, the difference in EV between the 40 and the 100 pattern classification is minute, that's why we added "although the improvement of EV seems to level off for high number of classes". But, we think in the presented context the statement is valid and we prefer to keep it as is.

Page 14 figure 7: I am amazed how relatively even the frequency distribution is. Expectation would be that some classes would be quite rare. Furthermore: is the property displayed really the quotient of frequency and cumulative EV? Furtherfurthermore, what does the (-) at the end of the y axis label mean?

Well, yes – frequency distribution ranges from 1.4 to 3.8. Which makes the most common pattern occurring more than twice as often as the rarest one, but that is still a rather narrow range. That behaviour originates from the optimisation algorithm used (SANDRA method). A classification using a leader algorithm would certainly result in a broader range of pattern frequencies.

Regarding the furthermores: I see that the y axis label caused some confusion. Of course the property displayed is not a quotient. The figure has been adapted.

Page 14 lines 30 and following as well as Fig. 8: It is remarkable and should be pointed out that for numerous classes the reanalyses (black dashes) mark either the lowest or the highest frequency so in those cases ALL GCMs are unanimously indicating either higher or lower frequencies, respectively. Isn't that an odd behavior?

Thanks for pointing this out. The described behaviour of GCMs (ALL being above or ALL being below the reanalysis) was observed for 7 patterns each. By having a closer look into this behaviour, it becomes apparent that particularly cold weather patterns (1, 12, 14, 21, 33, 34, 37) are underestimated, although the warm pattern 27 is also underestimated. Apparently, all GCMs have difficulties in reproducing these weather patterns. However, it goes beyond the scope of this manuscript to analyse the genesis of these weather patterns and why GCMs are not capable to capture them well. With regards to the overestimated patterns (3, 6, 7, 11, 20, 23, 35), they show a tendency towards average to above-average precipitation. But other, high precipitation patterns seem to be well-captured. The remaining 26 patterns enclose the reanalysis values in their range.

Page 14/15 Section 4.2.2 I assume that Fig. 9 on Page 16 is meant to visualize this, right? Then make a reference to that figure!

Thanks for that remark – the reference has been added.

Section 4.2.2 again: A definition needed is needed as to what is considered a *good* reproduction. Imagine that there could be ties in the months of most frequent occurrences - or months with very similar frequency. Would that still be good/superior/inferior reproductions, then?

We added the statement "A deviation of one month is considered an acceptably good performance." to give a definition for a good performance. A deviation of one month should be accepted as good performance, but since pattern occurrence stretches only across a couple of month, the peak should be matched +-1 month, everything else is not acceptable

Regarding ties (i.e. two months with an equal count of pattern occurrence) – this is observed only in pattern 9, model CNRM-CM5 where peak months are Jan and Dec. The resulting peak is marked in the middle between these two months, i.e. at 0.5. Usually there is a clear peak of occurrence - with substantially lower counts to the left and right of the peak month.

Page 15 Section 4.2.3 The text points to Figure 8, wheras the reference should point to Figure 10.

Thanks for pointing out the wrong reference – it has been corrected.

Page 18 around line 35, the aspect of stratification skill was presented in Spekat et al. (2010), too. Perhaps this needs to be mentioned in the text.

We mentioned papers that use a similar evaluation metric as we do (usually EV) to compare values. Spekat et al. (2010) show the reduction of variance by using the geopotential height, which we do not use in our classification. However, the inclusion of temperature in the CEC-TC classification in Spekat et al. (2010) indeed delivers a clear seasonal stratification. This is exactly what is also achieved in our analysis, however, without an additional explicit seasonal separation of weather patterns a priori.

Page 18 line 12 (bottom): More like a comment - this discussion opens up a whole philosophical can of worms, i.e., universality versus optimization. Should the goal be to find a classification able to cover "everything but with a variable degree of fidelity" or should the goal be to find a classification that is region- and variable-specific, yet has a high skill? Maybe the authors could be drawn into discussing this for a bit, too?

Yes, the reviewer is right the classification should be rather site and purpose specific. This is also one of the conclusions in the COST733 report. To this end, we focus particularly on the Rhine basin, for which we attempt an end-to-end attribution of flood changes to changes in the greenhouse gas concentrations due to anthropogenic emissions. For this purpose, we intend to use a hydrological model driven by a set of climate parameters (P, T, radiation, humidity) produced by the weather generator. Hence, we require profound multi-variate stratification of weather patterns for all above-mentioned quantities.

General comment on the "Discussion" section: It is rather long (no criticism concerning the length, mind you) and could benefit from the insertion of subsections.

We follow the suggestion of the reviewer and separate the discussion into two blocks "on the 'optimal' classification and "on the GCM skill".

Page 34 Table 1: Is it "Runs" or "Run", i.e., did the authors use all 10 CNRM runs or all 10 CSIRO runs (for example) or did they use just a specific one of those? Then this particular run should be specified. This refers back to page 3 line 12 where it is ambiguous if ALL or SOME runs were analyzed in this paper.

We used ALL available runs for the analyses. To make that point more clear, we added "Results from different runs of each GCM are averaged." in chapter 4.2 and changed the last sentence in chapter 2 to "All available runs were analysed…".

General comment concerning Figs. 11 thru 16: It is amazing to me that those 40 classes, some of which are *visually* quite similar to each other, apparently constitute sets of necessary distinctiveness. Just from looking at them the, admittedly subjective, estimate would be that much fewer classes should be sufficient.

The visual similarity of some classes is certainly correct. We partly cover that issue in chapter 3.3 and figure 3, concerning multi-variate evaluation. Nevertheless, some patterns seem to be quite similar across all variables (e.g. patterns 28 and 39). But it should be noted, that figures 11-16 show only the associated local variables, not the variables used for classification. They exhibit some differences in the underlying pattern, i.e. some patterns, although being different, generate similar local mean weather.

We tested a classification with 36 classes as well (though not shown in the manuscript). The results align closely to the other classifications (i.e. EV are slightly lower than for 40 classes).

General comment on the line numbering: It should be uninterrupted, starting with 1 and end in the high several hundreds. The numbering in the draft here is misguided and misguiding.

See answer to fifth comment. HESS' latex template apparently starts from 1 on each page again.

General comment on the figure placement: Particularly for Figs. 7 thru 10, a better proximity to their mentioning in the text and respective paragraphs to which they belong should be found.

Again, this is latex-specific and would anyway be adapted for the final production of a publication paper. Nevertheless, the figure placement has been adapted slightly.

General comment on Literature – one could of course think of a "me too" effect… - but there is a paper from 2010 which covers or complements several aspects of the manuscripts's reasoning: Spekat, A., F. Kreienkamp and W. Enke, 2010: An impact-oriented classification method for atmospheric patterns. – Physics and Chemistry of the Earth **35**, 352-359. Also, perhaps unknown to the authors of the manuscript, the final report for the COST733 Action is now available. The link to the final report is: https://opus.bibliothek.uni-augsburg.de/opus4/frontdoor/index/index/docId/3768

Thanks for giving these references. Both have been covered in the other comments already.

That link is permanent. The URN is urn:nbn:de:bvb:384-opus4-37682 (it can be found using a web search engine).

So, bottom line: Something in between minor and major revision. Some reasoning needs to be better shaped, some needs to mention a bit more what alternative paths have been pursued. There is some potential to improve technical aspects (figures, mostly) and general understandability.

We thank to the anonymous reviewer for the comments and careful examination of the manuscript, especially of the figures. This helped to improve some details using a new perspective.

[revised manuscript text omitted]

---

## Referee Comment (RC2) · Anonymous Referee #2 · 16 Aug 2016

Main impression: The paper presents an evaluation of downscaling climate information over the Rhine region by making use of weather patterns. A number of different options are explored, and the conclusion is that best results are obtained with mixed predictors: sea-level pressure in addition to temperature and humidity fields. The main new aspects of this study include the specific emphasis on the Rhine region, large number of stations representing the local climate, and the long time series for searching for weather patterns.

One question I have with this analysis is whether the evaluation of the method is best done when making use of cross-validation of split-sample for calibration/evaluation.

Perhaps the main message has a tendency to get lost in all the details? This could

be fixed with some revision and an emphasis/reminder of how the details support the main message. I think the abstract may be rewritten - a bit bolder - to make the paper look more interesting.

Details:

L15p2: "statistical approaches are comparatively cheap, computationally efficient and relatively easy to apply...". No, it is not always easy to apply statistical downscaling in a good fashion that correctly captures the dependency to large-scale conditions. However, it's easy to apply both dynamical and statistical downscaling to get some output - be it reliable results or non-representative numbers.

L28p2: "The underlying assumption of the downscaling based on weather patterns is that the regional or local behaviour of climate variables is a response to the larger-scale, synoptic forcing." More precisely, downscaling also works if only a fraction $f(X)$ of the variability (which one would expect) is dependent on the large-scale conditions $X$ (local processes $n$ are also usually involved): $y = f(X) + n$. However, both large-scale dependent and local variability must be accounted for. One case in point is precipitation, as discussed in the paper.

L31p2: "Statistical downscaling tends to underestimate the variance of regional or local climate and may poorly represent extremes". Some past studies have not accounted for the contribution local processes $n$, hence the variance in the results will be less than observed. Variance inflation is flawed and a priori gives incorrect results (von Stoch, 1999).

L12p3: The assumption of stationarity is more severe for GCMs and RCMs, which rely on parameterisation schemes, involving statistically trained equation to represent the bulk description of unresolved quantities (e.g. cloud schemes). In GCMs/RCMs the results of such schemes feed back to the calculation of the large-scales, whereas for statistical downscaling/weather generators, they can produce a trend in biases. Also relevant for L15p19. When mentioning this only in relation with statistical downscaling

(SD), the reader gets a distorted picture and thinks it only affects SD - this has resulted in a myth within the downscaling community.

"Data:" Gridded observation (EOBS) and station data were mixed? This can introduce artifacts (spatial and temporal inhomogeneities). Furthermore, gridded daily precipitation is no good for analysing extreme precipitation, as the grid points are weighted sums of surrounding observations and hence are expected to exhibit different statistical characteristics (tail of distribution - see attached fig).

Also see http://www.icrc-cordex2016.org/images/pdf/Programme/presentations/parallel_D/D3_Chandler_CORDEX2016.pdf Furthermore, isn't EOBS limited to after 1950? Perhaps it's better to skip France altogether even if the picture is less complete?

"3.1 Weather pattern classification" - the paper discloses that the cost733 class software was applied to both reanalysis and GCM data (?) - but does that mean that the weather patterns are the same for the models and reanalyses?

"3.2 Finding optimal classification parameters" Keep in mind that with many tests, the likelihood of finding an accidental match increases. The "problem of multiplicity" - See Wilks (2006).

Eq. 3 - 5: Daily rainfall amount is far from normally distributed, whereas the root-mean-square metric is more appropriate for temperature, which tends to behave more like the normal distribution. TSS, WSS and BSS will be strongly affected by a few heavy precipitation events (acknowledged in 4.1.2), and explains low scores for the metric EV. For precipitation, it may be wise to look at aggregated statistics, eg seasonal wet-day mean (precipitation intensity), wet-day frequency, and probabilities (e.g. Benestad & Mezghani, 2015): The precipitation frequency exhibits a close connection with the circulation pattern (e.g. SLP), whereas the intensity is more complicated and is expected to be strongly affected by local small-scale processes (eg convection, which may be consistent with Fig 7 and mentioned in the discussion), but be somewhat moderated by large-scale conditions. Furthermore, the observations represent a poor sample - a rain

gauge represents a few cm2 capture of a spatially heterogeneous phenomenon with a scale of km2. Aggregation in time or space may give a clearer picture that is less affected by sampling fluctuations. The alternative to downscaling single station data and then estimate the area average is to estimate the area average from observations and then downscale this index. I suggest adding some text about this possibility and these issues in the discussion, at least.

L3p10: The humidity estimate from reanalyses is difficult to validate - it may have substantial errors?

"4.2.2 Seasonality" - it's not clear what "the earliest and last months of occurrence in the course of the year" are and how they are specified.

"4.2.3 Persistence" - the duration of phenomenon/event/pattern may follow the geometric distribution, and differences in the models and reanalysis can be gauged based on its statistics. It can provide an estimate of what differences one would expect from randomness and what is likely a systematic bias.

Minor:

I would move the first sentence in the abstract to the beginning of the introduction. You don't need to explain why or provide justification in the abstract.

Second sentence in the abstract is a bit difficult, and can be rephrased or moved out of the abstract. It distracts the story away from the main findings. I'd start the abstract with "An objective classification scheme is presented . . ."

L28p4: "For the workflow proposed here three different sets of climate data are needed:" Comma between "here" and "three"?

L1p11: "The selected classification was compared to the Hess-Brezowsky-Grosswetterlagen" - Use "compared with" rather than "compared to" when there was an actual comparison?

[Figure]

References:

Storch, H. von. "On the Use of 'Inflation' in Statistical Downscaling." Journal of Climate 12 (1999): 3505–6. Wilks, D. S. "On 'Field Significance' and the False Discovery Rate." Journal of Applied Meteorology and Climatology 45, no. DOI: 10.1175/JAM2404.1 (2006): 1181–89. Benestad, Rasmus E., and Abdelkader Mezghani. "On Downscaling Probabilities for Heavy 24-Hour Precipitation Events at Seasonal-to-Decadal Scales." Tellus A 67, no. 0 (March 30, 2015). doi:10.3402/tellusa.v67.25954.
* * *
[Figure]

**Test gridded precip**

— Individual simulation
▬ Weighted sum

Frequency

amount
'gridded' result = weighted sum of 4 stations

**Fig. 1.**

---

## Author Comment (AC2) · 19 Aug 2016

*We thank the reviewer for his/her constructive comments on our manuscript and provide point-by-point response hereafter.*

Main impression: The paper presents an evaluation of downscaling climate information over the Rhine region by making use of weather patterns. A number of different options are explored, and the conclusion is that best results are obtained with mixed predictors: sea-level pressure in addition to temperature and humidity fields. The main new aspects of this study include the specific emphasis on the Rhine region, large number of stations representing the local climate, and the long time series for searching for weather patterns.

[Figure]

One question I have with this analysis is whether the evaluation of the method is best done when making use of cross-validation of split-sample for calibration/evaluation. Perhaps the main message has a tendency to get lost in all the details? This could be fixed with some revision and an emphasis/reminder of how the details support the main message. I think the abstract may be rewritten – a bit bolder – to make the paper look more interesting.

*With regards to the reviewers comment, we point out that the classification of weather patterns is done based on the ERA-20C reanalysis data, whereas the evaluation of the classifications' capability to stratify local climatological variables is evaluated using an independent data set based on climate station observations.*

*We shall consider the reviewer's suggestion and make the abstract more appealing pointing out our main messages.*

Details:

L15p2: "statistical approaches are comparatively cheap, computationally efficient and relatively easy to apply...". No, it is not always easy to apply statistical downscaling in a good fashion that correctly captures the dependency to large-scale conditions. However, it's easy to apply both dynamical and statistical downscaling to get some output – be it reliable results or non-representative numbers.

*We adapt the respective paragraph, deleting the criticised sentence.*

L28p2: "The underlying assumption of the downscaling based on weather patterns is that the regional or local behaviour of climate variables is a response to the larger-scale, synoptic forcing." More precisely, downscaling also works if only a fraction $f(X)$ of the variability (which one would expect) is dependent on the large-scale conditions $X$ (local processes $n$ are also usually involved): $y = f(X) + n$. However, both large-scale dependent and local variability must be accounted for. One case in point is precipitation, as discussed in the paper.

*We acknowledge the reviewers comment and would change the mentioned sentence to "The underlying assumption of the downscaling based on weather patterns is that the regional or local behaviour of climate variables is* partly *a response to the larger-scale, synoptic forcing."*

L31p2: "Statistical downscaling tends to underestimate the variance of regional or local climate and may poorly represent extremes". Some past studies have not accounted for the contribution local processes n, hence the variance in the results will be less than observed. Variance inflation is flawed and a priori gives incorrect results (von Stoch, 1999).

*We will adapt the statement accordingly: "Statistical downscaling tends to underestimate the variance of regional or local climate if the contribution of local processes is not considered and may poorly represent extremes. Different methods have been proposed to rectify this problem: variable inflation (Karl et al., 1990), expanded downscaling (Bürger, 1996) and randomisation (Kilsby et al., 1998)."*

L12p3: The assumption of stationarity is more severe for GCMs and RCMs, which rely on parameterisation schemes, involving statistically trained equation to represent the bulk description of unresolved quantities (e.g. cloud schemes). In GCMs/RCMs the results of such schemes feed back to the calculation of the large-scales, whereas for statistical downscaling/weather generators, they can produce a trend in biases. Also relevant for L15p19. When mentioning this only in relation with statistical downscaling (SD), the reader gets a distorted picture and thinks it only affects SD – this has resulted in a myth within the downscaling community.

*Yes, we agree with the reviewer that dynamical downscaling techniques also rely on the assumptions of stationarity in the empirical relationships incorporated in the climate models. In our study we, however, solely focus on the statistical downscaling and even more narrowly on the weather generator type downscaling conditioned on weather patterns. We thus discuss only the assumptions required for this downscaling*

*approach.*

"Data:" Gridded observation (EOBS) and station data were mixed? This can introduce artifacts (spatial and temporal inhomogeneities). Furthermore, gridded daily precipitation is no good for analysing extreme precipitation, as the grid points are weighted sums of surrounding observations and hence are expected to exhibit different statistical characteristics (tail of distribution – see attached fig). Also see http://www.icrc-cordex2016.org/images/pdf/Programme/presentations/parallel_D/D3_Chandler_C

Furthermore, isn't EOBS limited to after 1950? Perhaps it's better to skip France altogether even if the picture is less complete?

*The reviewer raised valid concerns regarding the use of E-OBS data. We decided to exclude these data points from our analyses and update all figures and results accordingly. However, the influence of these data points is only minimal, thus the overall messages of the paper remain unchanged.*

"3.1 Weather pattern classification" - the paper discloses that the cost733 class software was applied to both reanalysis and GCM data (?) – but does that mean that the weather patterns are the same for the models and reanalyses?

*Apparently, our description of the workflow was misleading. We first establish a classification on reanalysis data. The GCM data are then assigned to the existing patterns (by means of minimum Euclidean distance). Thus, the weather patterns are the same for the reanalysis and the GCMs. We adapt the first sentence in section 4.2 accordingly.*

"3.2 Finding optimal classification parameters" Keep in mind that with many tests, the likelihood of finding an accidental match increases. The "problem of multiplicity" – See Wilks (2006).

*Multiplicity applies to statistical tests. We do not apply multiple tests, but rather evaluate a set of different valid parameters by using two different metrics. Hence, we do not share the concerns of the reviewer.*

Eq. 3 – 5: Daily rainfall amount is far from normally distributed, whereas the root-mean-square metric is more appropriate for temperature, which tends to behave more like the normal distribution. TSS, WSS and BSS will be strongly affected by a few heavy precipitation events (acknowledged in 4.1.2), and explains low scores for the metric EV. For precipitation, it may be wise to look at aggregated statistics, eg seasonal wet-day mean (precipitation intensity), wet-day frequency, and probabilities (e.g. Benestad & Mezghani, 2015): The precipitation frequency exhibits a close connection with the circulation pattern (e.g. SLP), whereas the intensity is more complicated and is expected to be strongly affected by local small-scale processes (eg convection, which may be consistent with Fig 7 and mentioned in the discussion), but be somewhat moderated by large-scale conditions.

*We thank the reviewer for the suggestion and will consider the mentioned characteristics (wet-day mean, wet-day frequency, and probabilities) in the revised version.*

Furthermore, the observations represent a poor sample – a rain gauge represents a few $cm^2$ capture of a spatially heterogeneous phenomenon with a scale of $km^2$. Aggregation in time or space may give a clearer picture that is less affected by sampling fluctuations. The alternative to downscaling single station data and then estimate the area average is to estimate the area average from observations and then downscale this index. I suggest adding some text about this possibility and these issues in the discussion, at least.

*We share the concern of the reviewer that station records represent only a point estimate of the spatially heterogeneous phenomenon like precipitation. We, however, intend to apply an existing advanced weather generator from Hundecha et al. (2009) which works on a station basis. Hence, we would abstain from interpolation of precipitation and downscaling regionalized indices instead of estimations at station locations.*

L3p10: The humidity estimate from reanalyses is difficult to validate – it may have substantial errors?

*We generally agree. In the relatively short overlapping period, however, ERA-20C moisture variables such as precipitation and total column water vapour agree fairly well with observations (cf. Poli et al. (2016), DOI: 10.1175/JCLI-D-15-0556.1). Besides, our study can also be viewed as a validation study itself.*

"4.2.2 Seasonality" - it's not clear what "the earliest and last months of occurrence in the course of the year" are and how they are specified.

*We add another paragraph in the end of section 4.2 to explain more thoroughly, what we mean by seasonality of patterns: "Seasonality is evaluated by the first, last, and peak month of pattern occurrence. All patterns show a distinct seasonality. Each season is characterised by a limited number of consecutive months in which a pattern occurs. We evaluate the beginning (i.e. first month) and end (i.e. last month) of pattern occurrence. The peak month is defined as the month with highest number of days with pattern occurrence. Some patterns show two distinct seasons. In this case both seasons are evaluated separately."*

"4.2.3 Persistence" - the duration of phenomenon/event/pattern may follow the geometric distribution, and differences in the models and reanalysis can be gauged based on its statistics. It can provide an estimate of what differences one would expect from randomness and what is likely a systematic bias.

*Our point here was solely a comparison of duration times and not the question whether there is any persistence at all. For that one could use the geometric distribution, but it would constitute a rather weak null hypothesis since in that case consecutive days are treated independently, which they are obviously not.*

Minor:

I would move the first sentence in the abstract to the beginning of the introduction. You don't need to explain why or provide justification in the abstract.

*To move the first sentence from the abstract to Introduction would make this sentence*

[Figure]

*stand-alone followed by the description of the basin and the flood problem. In the abstract, it provides an overall idea of the scope of the study. We would prefer to keep the current structure, if the reviewer agrees.*

Second sentence in the abstract is a bit difficult, and can be rephrased or moved out of the abstract. It distracts the story away from the main findings. I'd start the abstract with "An objective classification scheme is presented . . ."

*Thanks for pointing this out. We shall critically revise the abstract, in particular the first statements and in general make it clearer and 'bolder' as suggested by the reviewer earlier.*

L28p4: "For the workflow proposed here three different sets of climate data are needed:" Comma between "here" and "three"?

L1p11: "The selected classification was compared to the Hess-Brezowsky- Grosswet-terlagen" - Use "compared with" rather than "compared to" when there was an actual comparison?

*Thanks, the corrections will be done.*

---

## Author Response (AR1)

**Authors' response**

**We thank the reviewers for their constructive comments on our manuscript and provide point-by-point response hereafter.**

**Reviewer 1:**

There was an initial reaction upon studying the manuscript: 40 patterns?? Are you serious??

**Answer:**

Yes, we are. Number 40 for the number of classes is not unusual (Philipp et al., 2009). We give various reasons in the discussion and in chapter 3.2 for choosing this number of classes. Among others, we need high stratification of variables for a multi-variate weather generator, which will be subsequently used for global model downscaling. This is best approached by a high number of classes. The availability of a long daily time series of daily climate observations (111 years) in the Rhine catchment further justifies our selection.

**R1:**

Then another question arose: Are pressure fields as classification a good basis for the stratification of data? There is good experience with using relative topography instead (Spekat et al., 2010 – reference is given at the end).

**Answer:**

No doubt, there are other suitable variables to be used for a classification besides pressure fields. By the way, our 'optimal' classification is based on a combination of sea level pressure, temperature and humidity. As we stated in chapter 3.2: "However our selection was restricted to variables that are also available from the GCM outputs." Since the classification is to be used for an attribution study later on, we need to select variables for classification that are available for "historical" and "historicalNat" runs of the CMIP5 project. To date geopotential height is only available for 3 Models (10 runs in total) of the historicalNat experiment and 9 Models of the historical experiment. Thus, we would reduce our data base considerably if we decided to use geopotential height or any parameter derived from that. Geopotential height is related to temperature and pressure. We show in Fig. 6 that a classification that uses geopotential height (among other variables) does not perform better than our "optimal" classification, which includes mean sea level pressure and temperature.

**R1:**

The authors do not go into seasonality when analyzing their data. There is contrary experience from the COST733 Action on classification methods (in the meantime, the results of that COST Action have been made publicly available, see reference at the end). Plus there is experience towards great usefulness of seasonality in classification, indicated, e.g., in Spekat et al. (2010).

**Answer:**

We do mention (and evaluate) the seasonality of our patterns (chapter 4.1.1 p 10). We show a distinct pattern seasonality due to the use of temperature as classification variable. By using absolute temperature (not anomalies) and a relatively large number of patterns, we achieve seasonal pattern stratification. We do not establish one classification per season, i.e. we "allow" our patterns to assign to their respective season by themselves (and to move within the year under the effect of climate change, if necessary). To make it short: we account for seasonality implicitly (in contrast to other studies that use one separate classification per pre-defined season). Thanks for pointing out to the COST733 Action final report. Many (most) results of the report have been already published some years ago and we are well aware of them (many being cited in the manuscript).

**R1:**

Usage of E-OBS data – Fig. 1 only shows 8 E-OBS grid points. Text mentions 10. Moreover in Fig. 1: Poor choice of colours for dots.

**Answer:**

We decided to exclude these data points from our analyses, following reviewer 2, and updated all figures and results accordingly.

**R1:**

From page 5 on the line numbering is irritatingly confusing. Just look at the repetition of line number 5 on page 5...

**Answer:**

True... the manuscript has been compiled using HESS' Latex template. Apparently something went wrong. It is fixed now.

**R1:**

Section 3.2 page 7 line 12: Maybe relative topography is not directly available, but contributing geopotentials can be easily extracted and the retop could be easily computed.

**Answer:**

See answer to the second comment.

**R1:**

Page 7 line 14 – and I mean the second appearance of this line (sigh...) beginngin with "(extents" it must be made more clear that these numbers refer to geographic degrees of latitude and longitude.

**Answer:**

This has been adapted now in the text (last paragraph of 3.2) and in the caption of figure 2.

**R1:**

Page 10 figure 3: Not clear what the numbers "12" , "14" and "33" on the right-hand side of the array of figures mean. I would furthermore recommend to deliberately use different colour schemes for different parameters, so they can be better distinguished. Even if the authors would not follow this suggestion at least they should reverse the assignment of the colours (left side red, right side blue) for PREC and HUMID since blue would then point to wet condition which suits the intuition better. More a thought than a substantial comment: All selected patterns are rather cold/have rather low radiation, so perhaps one would like to see examples for classes which denote different conditions.

**Answer:**

The numbers 12, 14, and 33 indicate the number of the pattern that have been selected. This has been pointed out now in the text (second last paragraph of 3.3) and in the caption of figure 3. The colour scales have been adapted. Regarding the comment concerning pattern selection: the point of this selection is to show patterns that are very similar in some variables (in this case temperature and radiation), but differ in other variables (here: precipitation and humidity). This aims at pointing out the need of multi-variate classification as we describe in the second last paragraph of 3.3.

**R1:**

Page 10 line 8 I have doubts if retaining the absolute values is a good approach when you have season-specific classes. Particularly temperature is way different from season to season. So au contraire to what the authors write, anomalies are a good way forward because they cover a similar value range in all seasons. Moreover, there is the experience that a further reduction in the number of classes is possible by using anomalies (this would be favourable in the light of the big set of 40 classes used by the authors).

**Answer:**

We would like to refer the reviewer to our answer to his/her third comment. We do not define the patterns explicitly for each season. This is basically achieved through using the absolute values for temperature as classification variable. The advantage is that we have a continuous sequence of patterns. The fact that each pattern occurs only in a rather limited number of months is solely due to the usage of absolute temperature values for classification. We do know that anomalies are used in many other papers and we are well aware that the annual signal of temperature dominates our classification. See e.g. second paragraph of 4.1.1. Nevertheless, regarding the future use of that classification (downscaling tool in climate change attribution), we appreciate the seasonal restriction of patterns a lot.

**R1:**

Page 11 figure 4: I suggest to add y-axis labels on the right-hand side, too. Further suggestion: Use open triangle and open circles which are better visible in case of overlaps, and those do appear frequently.

**Answer:**

The figures have been updated using open symbols now. However, we would like to refrain from adding a second y axis. The grid lines in the plots should be sufficient to guide the reader's eye.

**R1:**

Page 12 figure 5: It would be could to have the results for four different class number shown at least for one more extent (or domain size, as one might better call it).

**Answer:**

We would prefer to refrain from that as well. The parameter selection follows a clear strategy, as we point out in the last paragraph of 3.2: 1. Select variable, 2. Select spatial domain, 3. Select number of classes. Adding further information into the figures would jeopardize its clarity.

**R1:**

Page 12 line 14: "increasing EV" – this is very minute if you look at it. Therefore I suggest to write "almost no change".

**Answer:**

The mentioned paragraph states: "(...) confirming the general tendency (increasing EV, decreasing PF values for increasing number of classes), although the improvement of EV seems to level off for high number of classes, meaning that the gain in stratification skill is only minimal." – Increasing tendency refers to the fact that a higher number of classes results in higher EV values (which is evident from figure 5). Of course, as the reviewer has mentioned, the difference in EV between the 40 and the 100 pattern classification is minute, that's why we added "although the improvement of EV seems to level off for high number of classes". But, we think in the presented context the statement is valid and we prefer to keep it as is.

**R1:**

Page 14 figure 7: I am amazed how relatively even the frequency distribution is. Expectation would be that some classes would be quite rare. Furthermore: is the property displayed really the quotient of frequency and cumulative EV? Furtherfurthermore, what does the (-) at the end of the y axis label mean?

**Answer:**

Well, yes – frequency distribution ranges from 1.4 to 3.8. Which makes the most common pattern occurring almost three times as often as the rarest one, but that is still a rather narrow range. That behaviour originates from the optimisation algorithm used (SANDRA method). A classification using a leader algorithm would certainly result in a broader range of pattern frequencies. Regarding the furthermores: I see that the y axis label caused some confusion. Of course the property displayed is not a quotient. The figure has been adapted.

**R1:**

Page 14 lines 30 and following as well as Fig. 8: It is remarkable and should be pointed out that for numerous classes the reanalyses (black dashes) mark either the lowest or the highest frequency so in those cases ALL GCMs are unanimously indicating either higher or lower frequencies, respectively. Isn't that an odd behavior?

**Answer:**

Thanks for pointing this out. The described behaviour of GCMs (ALL being above or ALL being below the reanalysis) was observed for 7 patterns each. By having a closer look into this behaviour, it becomes apparent that particularly cold weather patterns (1, 12, 14, 21, 33, 34, 37) are underestimated, although the warm pattern 27 is also underestimated. Apparently, all GCMs have difficulties in reproducing these weather patterns. However, it goes beyond the scope of this manuscript to analyse the genesis of these weather patterns and why GCMs are not capable to capture them well. With regards to the overestimated patterns (3, 6, 7, 11, 20, 23, 35), they show a tendency towards average to above-average precipitation. But other, high precipitation patterns seem to be well-captured. The remaining 26 patterns enclose the reanalysis values in their range..

**R1:**

Page 14/15 Section 4.2.2 I assume that Fig. 9 on Page 16 is meant to visualize this, right? Then make a reference to that figure!

**Answer:**

Thanks for that remark – the reference has been added.

**R1:**

Section 4.2.2 again: A definition needed is needed as to what is considered a good reproduction. Imagine that there could be ties in the months of most frequent occurrences – or months with very similar frequency. Would that still be good/superior/inferior reproductions, then?

**Answer:**

We added the statement "A deviation of one month is considered an acceptably good performance." to give a definition for a good performance. A deviation of one month should be accepted as good performance, but since pattern occurrence stretches only across a couple of month, the peak should be matched +-1 month, everything else is not acceptable Regarding ties (i.e. two months with an equal count of pattern occurrence) – this is observed only in pattern 9, model CNRM-CM5 where peak months are Jan and Dec. The resulting peak is marked in the middle between these two months, i.e. at 0.5. Usually there is a clear peak of occurrence – with substantially lower counts to the left and right of the peak month.

**R1:**

Page 15 Section 4.2.3 The text points to Figure 8, wheras the reference should point to Figure 10.

**Answer:**

Thanks for pointing out the wrong reference – it has been corrected.

**R1:**

Page 18 around line 35, the aspect of stratification skill was presented in Spekat et al. (2010), too. Perhaps this needs to be mentioned in the text.

**Answer:**

We mentioned papers that use a similar evaluation metric as we do (usually EV) to compare values. Spekat et al. (2010) show the reduction of variance by using the geopotential height, which we do not use in our classification. However, the inclusion of temperature in the CEC-TC classification in Spekat et al. (2010) indeed delivers a clear seasonal stratification. This is exactly what is also achieved in our analysis, however, without an additional explicit seasonal separation of weather patterns a priori.

**R1:**

Page 18 line 12 (bottom): More like a comment – this discussion opens up a whole philosophical can of worms, i.e., universality versus optimization. Should the goal be to find a classification able to cover "everything but with a variable degree of fidelity" or should the goal be to find a classification that is region- and variable-specific, yet has a high skill? Maybe the authors could be drawn into discussing this for a bit, too?

**Answer:**

Yes, the reviewer is right the classification should be rather site and purpose specific. This is also one of the conclusions in the COST733 report. To this end, we focus particularly on the Rhine basin, for which we attempt an end-to-end attribution of flood changes to changes in the greenhouse gas concentrations due to anthropogenic emissions. For this purpose, we intend to use a hydrological model driven by a set of climate parameters (P,T, radiation, humidity) produced by the weather generator. Hence, we require profound multi-variate stratification of of weather patterns for all above-mentioned quantities.

**R1:**

General comment on the "Discussion" section: It is rather long (no criticism concerning the length, mind you) and could benefit from the insertion of subsections.

**Answer:**

We follow the suggestion of the reviewer and separate the discussion into two blocks "on the 'optimal' classification" and "on the GCM skill"

**R1:**

Page 34 Table 1: Is it "Runs" or "Run", i.e., did the authors use all 10 CNRM runs or all 10 CSIRO runs (for example) or did they use just a specific one of those? Then this particular run should be specified. This refers back to page 3 line 12 where it is ambiguous if ALL or SOME runs were analyzed in this paper.

**Answer:**

We used ALL available runs for the analyses. To make that point more clear, we added "Results from different runs of each GCM are averaged." in chapter 4.2 and changed the last sentence in chapter 2 to "All available runs were analysed?".

**R1:**

General comment concerning Figs. 11 thru 16: It is amazing to me that those 40 classes, some of which are visually quite similar to each other, apparently constitute sets of necessary distinctiveness. Just from looking at them the, admittedly subjective, estimate would be that much fewer classes should be sufficient.

**Answer:**

The visual similarity of some classes is certainly correct. We partly cover that issue in chapter 3.3 and figure 3, concerning multi-variate evaluation. Nevertheless, some patterns seem to be quite similar across all variables (e.g. patterns 28 and 39). But it should be noted, that figures 11-16 show only the associated local variables, not the variables used for classification. They exhibit some differences in the underlying pattern, i.e. some patterns, although being different, generate similar local mean weather. We tested a classification with 36 classes as well (though not shown in the manuscript). The results align closely to the other classifications (i.e. EV are slightly lower than for 40 classes).

**R1:**

General comment on the line numbering: It should be uninterrupted, starting with 1 and end in the high several hundreds. The numbering in the draft here is misguided and misguiding.

**Answer:**

See answer to fifth comment. HESS' latex template apparently starts from 1 on each page again.

**R1:**

General comment on the figure placement: Particularly for Figs. 7 thru 10, a better proximity to their mentioning in the text and respective paragraphs to which they belong should be found.

**Answer:**

Again, this is latex-specific and would anyway be adapted for the final production of a publication paper. Nevertheless, the figure placement has been adapted slightly.

**R1:**

General comment on Literature – one could of course think of a "me too" effect? – but there is a paper from 2010 which covers or complements several aspects of the manuscripts?s reasoning: Spekat, A., F. Kreienkamp and W. Enke, 2010: An impact-oriented classification method for atmospheric patterns. – Physics and Chemistry of the Earth 35, 352-359. Also, perhaps unknown to the authors of the manuscript, the final report for the COST733 Action is now available. The link to the final report is: https://opus.bibliothek.uni-augsburg.de/opus4/frontdoor/index/index/docId/3768 That link is permanent. The URN is urn:nbn:de:bvb:384-opus4-37682 (it can be found using a web search engine).

**Answer:**

Thanks for giving these references. Both have been covered in the other comments already.

**R1:**

So, bottom line: Something in between minor and major revision. Some reasoning needs to be better shaped, some needs to mention a bit more what alternative paths have been pursued. There is some potential to improve technical aspects (figures, mostly) and general understandability.

**Answer:**

We thank to the anonymous reviewer for the comments and careful examination of the manuscript, especially of the figures. This helped to improve some details using a new perspective.

**Reviewer 2:**

Main impression: The paper presents an evaluation of downscaling climate information over the Rhine region by making use of weather patterns. A number of different options are explored, and the conclusion is that best results are obtained with mixed predictors: sea-level pressure in addition to temperature and humidity fields. The main new aspects of this study include the specific emphasis on the Rhine region, large number of stations representing the local climate, and the long time series for searching for weather patterns.

One question I have with this analysis is whether the evaluation of the method is best done when making use of cross-validation of split-sample for calibration/evaluation. Perhaps the main message has a tendency to get lost in all the details? This could be fixed with some revision and an emphasis/reminder of how the details support the main message. I think the abstract may be rewritten – a bit bolder – to make the paper look more interesting.

**Answer:**

With regards to the reviewers comment, we point out that the classification of weather patterns is done based on the ERA-20C reanalysis data, whereas the evaluation of the classifications' capability to stratify local climatological variables is evaluated using an independent data set based on climate station observations.

We re-wrote the abstract to meet the reviewer's suggestion.

**R2:**

L15p2: "statistical approaches are comparatively cheap, computationally efficient and relatively easy to apply?". No, it is not always easy to apply statistical downscaling in a good fashion that correctly captures the dependency to large-scale conditions. However, it?s easy to apply both dynamical and statistical downscaling to get some output – be it reliable results or non-representative numbers.

**Answer:**

We adapted the respective paragraph, deleting the criticised sentence.

**R2:**

L28p2: "The underlying assumption of the downscaling based on weather patterns is that the regional or local behaviour of climate variables is a response to the larger-scale, synoptic forcing." More precisely, downscaling also works if only a fraction $f(X)$ of the variability (which one would expect) is dependent on the large-scale conditions $X$ (local processes $n$ are also usually involved): $y = f(X) + n$. However, both large-scale dependent and local variability must be accounted for. One case in point is precipitation, as discussed in the paper.

**Answer:**

We acknowledge the reviewers comment and changed the mentioned sentence to "The underlying assumption of the downscaling based on weather patterns is that the regional or local behaviour of climate variables is *partly* a response to the larger-scale, synoptic forcing."

**R2:**

L31p2: "Statistical downscaling tends to underestimate the variance of regional or local climate and may poorly represent extremes". Some past studies have not accounted for the contribution local processes n, hence the variance in the results will be less than observed. Variance inflation is flawed and a priori gives incorrect results (von Stoch, 1999).

**Answer:**

We adapted the statement accordingly: "Statistical downscaling tends to underestimate the variance of regional or local climate if the contribution of local processes is not considered and may poorly represent extremes. Different methods have been proposed to rectify this problem: variable inflation (Karl et al., 1990), expanded downscaling (Bürger, 1996) and randomisation (Kilsby et al., 1998)."

**R2:**

L12p3: The assumption of stationarity is more severe for GCMs and RCMs, which rely on parameterisation schemes, involving statistically trained equation to represent the bulk description of unresolved quantities (e.g. cloud schemes). In GCMs/RCMs the results of such schemes feed back to the calculation of the large-scales, whereas for statistical downscaling/weather generators, they can produce a trend in biases. Also relevant for L15p19. When mentioning this only in relation with statistical downscaling (SD), the reader gets a distorted picture and thinks it only affects SD – this has resulted in a myth within the downscaling community.

**Answer:**

Yes, we agree with the reviewer that dynamical downscaling techniques also rely on the assumptions of stationarity in the empirical relationships incorporated in the climate models. In our study we, however, solely focus on the statistical downscaling and even more narrowly on the weather generator type downscaling conditioned on weather patterns. We thus discuss only the assumptions required for this downscaling approach.

**R2:**

"Data:" Gridded observation (EOBS) and station data were mixed? This can introduce artifacts (spatial and temporal inhomogeneities). Furthermore, gridded daily precipitation is no good for analysing extreme precipitation, as the grid points are weighted sums of surrounding observations and hence are expected to exhibit different statistical characteristics (tail of distribution – see attached fig). Also see http://www.icrc-cordex2016.org/images/pdf/Programme/presentations/parallel_D/D3_Chandler_CORDEX2016.pdf.
   Furthermore, isn't EOBS limited to after 1950? Perhaps it's better to skip France altogether even if the picture is less complete?

**Answer:**

The reviewer raised valid concerns regarding the use of E-OBS data. We decided to exclude these data points from our analyses and updated all figures and results accordingly. However, the influence of these data points is only minimal, thus the overall messages of the paper remain unchanged.

**R2:**

"3.1 Weather pattern classification" – the paper discloses that the cost733 class software was applied to both reanalysis and GCM data (?) – but does that mean that the weather patterns are the same for the models and reanalyses?

**Answer:**

Apparently, our description of the workflow was misleading. We first establish a classification on reanalysis data. The GCM data are then assigned to the existing patterns (by means of minimum Euclidean distance). Thus, the weather patterns are the same for the reanalysis and the GCMs. We adapted the first sentence in section 4.2 accordingly.

**R2:**

"3.2 Finding optimal classification parameters" Keep in mind that with many tests, the likelihood of finding an accidental match increases. The "problem of multiplicity" – See Wilks (2006).

**Answer:**

Multiplicity applies to statistical tests. We do not apply multiple tests, but rather evaluate a set of different valid parameters by using two different metrics. Hence, we do not share the concerns of the reviewer.

**R2:**

Eq. 3 – 5: Daily rainfall amount is far from normally distributed, whereas the root-mean-square metric is more appropriate for temperature, which tends to behave more like the normal distribution. TSS, WSS and BSS will be strongly affected by a few heavy precipitation events (acknowledged in 4.1.2), and explains low scores for the metric EV. For precipitation, it may be wise to look at aggregated statistics, eg seasonal wet-day mean (precipitation intensity), wet-day frequency, and probabilities (e.g. Benestad & Mezghani, 2015): The precipitation frequency exhibits a close connection with the circulation pattern (e.g. SLP), whereas the intensity is more complicated and is expected to be strongly affected by local small-scale processes (eg convection, which may be consistent with Fig 7 and mentioned in the discussion), but be somewhat moderated by large-scale conditions.

**Answer:**

Following the reviewer's suggestion we analysed wet day frequency and wet day mean precipitation (i.e. intensity) for all seasons and the whole year. While precipitation frequency differs between patterns (between <10% for e.g. pattern 36 and >90% for pattern 39), the variation among patterns is much smaller for precipitation intensity. Patterns with high frequencies are associated with high mean daily precipitation (as in Figure 12). Frequencies and intensities differ only slightly between seasons.

   We conclude that the differences between pattern precipitation (as in Figure 12) are due to differences in precipitation frequency rather than intensity. We added a respective paragraph to the end of section 4.1.3, but did not add any extra figures since they do not contain additional information.

**R2:**

Furthermore, the observations represent a poor sample – a rain gauge represents a few $cm^2$ capture of a spatially heterogeneous phenomenon with a scale of $km^2$. Aggregation in time or space may give a clearer picture that is less affected by sampling fluctuations. The alternative to downscaling single station data and then estimate the area average is to estimate the area average from observations and then downscale this index. I suggest adding some text about this possibility and these issues in the discussion, at least.

**Answer:**

We share the concern of the reviewer that station records represent only a point estimate of the spatially heterogeneous phenomenon like precipitation. We, however, intend to apply an existing advanced weather generator from Hundecha et al. (2009) which works on a station basis. Hence, we would abstain from interpolation of precipitation and downscaling regionalized indices instead of estimations at station locations.

**R2:**

L3p10: The humidity estimate from reanalyses is difficult to validate – it may have substantial errors?

**Answer:**

We generally agree. In the relatively short overlapping period, however, ERA-20C moisture variables such as precipitation and total column water vapour agree fairly well with observations (cf. Poli et al. (2016), DOI: 10.1175/JCLI-D-15-0556.1). Besides, our study can also be viewed as a validation study itself.

**R2:**

"4.2.2 Seasonality" – it's not clear what "the earliest and last months of occurrence in the course of the year" are and how they are specified.

**Answer:**

We added another paragraph in the end of section 4.2 to explain more thoroughly, what we mean by seasonality of patterns: "Seasonality is evaluated by the first, last, and peak month of pattern occurrence. All patterns show a distinct seasonality. Each season is characterised by a limited number of consecutive months in which a pattern occurs. We evaluate the beginning (i.e. first month) and end (i.e. last month) of pattern occurrence. The peak month is defined as the month with highest number of days with pattern occurrence. Some patterns show two distinct seasons. In this case both seasons are evaluated separately."

**R2:**

"4.2.3 Persistence" - the duration of phenomenon/event/pattern may follow the geometric distribution, and differences in the models and reanalysis can be gauged based on its statistics. It can provide an estimate of what differences one would expect from randomness and what is likely a systematic bias.

**Answer:**

Our point here was solely a comparison of duration times and not the question whether there is any persistence at all. For that one could use the geometric distribution, but it would constitute a rather weak null hypothesis since in that case consecutive days are treated independently, which they are obviously not.

**R2:**

Minor:
   I would move the first sentence in the abstract to the beginning of the introduction. You don't need to explain why or provide justification in the abstract.
   Second sentence in the abstract is a bit difficult, and can be rephrased or moved out of the abstract. It distracts the story away from the main findings. I'd start the abstract with "An objective classification scheme is presented ?"

**Answer:**

To move the first sentence from the abstract to Introduction would make this sentence stand-alone followed by the description of the basin and the flood problem. In the abstract, it provides an overall idea of the scope of the study. We would prefer to keep the current structure, but re-wrote parts of the abstract nonetheless as also required by the other reviewer.

**R2:**

L28p4: "For the workflow proposed here three different sets of climate data are needed:" Comma between "here" and "three"?

L1p11: "The selected classification was compared to the Hess-Brezowsky- Grosswetterlagen" - Use "compared with" rather than "compared to" when there was an actual comparison?

**Answer:**

Thanks, the corrections were done.

**Can local climate variability be explained by weather patterns? A multi-station evaluation for the Rhine basin**

Aline Murawski, Gerd Bürger, Sergiy Vorogushyn, Bruno Merz[1], Aline Murawski[1], Gerd Bürger[2, 3], Sergiy Vorogushyn[1], and Bruno Merz[1]

[1]GFZ German Research Centre for Geosciences, Potsdam, Germany
[2]Institute of Meteorology, FU Berlin, Germany
[3]Institute of Earth and Environmental Science, University of Potsdam, Germany

*Correspondence to:* Aline Murawski (murawski@gfz-potsdam.de)

**Abstract.** For understanding past flood changes in the Rhine catchment and in particular  the role of anthropogenic climate change for extreme flows, an attribution study relying on a proper GCM (General Circulation Model) downscaling is needed. A downscaling based on conditioning a stochastic weather generator on weather patterns is a promising approach  . This approach assumes a strong link between weather patterns and local climate, and sufficient GCM skill in reproducing weather pattern climatology.  These presuppositions are unprecedentedly evaluated here using 111 years of daily climate data from  490 stations in the Rhine basin  and comprehensively testing the number of classification parameters and GCM weather pattern characteristics. A classification based on a combination of mean sea level pressure, temperature, and humidity from the ERA20C reanalysis  of atmospheric fields over Central Europe with  40 weather  types was found the most appropriate to stratify six local climate variables. The  corresponding skill is quite diverse though, ranging from good for radiation to  poor for precipitation. Especially ~~local precipitation and humidity are governed by processes that are not completely represented by the large-scale distribution of pressure, temperature and humidity. Before applying the weather pattern based downscaling approach, it should therefore be investigated whether the link between the large-scale synoptic situation and the local climate variable of interest is strong enough for the given purpose. Our analysis suggests that it is advantageous to incorporate additional classification variables besides pressure fields . The use of temperature 
[revised manuscript text omitted]